

# A systematic review and meta-analysis of randomized controlled trials for physical activity among colorectal cancer survivors: directions for future research

Jiayu Mao[1], Xiaoke Qiu[1], Yi Zhang[1], Can Wang[1], Xueli Yang[2] and Qiuping Li[1,2]

[1] Wuxi School of Medicine, Jiangnan University, Wuxi, Jiangsu Province, China
[2] Affiliated Hospital of Jiangnan University, Wuxi, Jiangsu Province, China

Corresponding author
Qiuping Li, liqp@163.com

## ABSTRACT

**Background:** Physical activity (PA) is critically important to cancer rehabilitation. However, PA levels are generally lower in colorectal cancer (CRC) survivors compared to other cancer survivors. The purpose of this study was to examine the effectiveness of PA interventions in increasing PA levels and to provide recommendations for developing PA interventions in CRC survivors.

**Methods:** A systematic literature search was conducted in Cochrane Library, Embase, PubMed, Scopus, Web of Science, China National Knowledge Infrastructure, and Wan Fang Data from January 2010 to March 1, 2024. The Physiotherapy Evidence Database (PEDro) scale was used to assess the methodological quality of eligible studies, and the Grading of Recommendations, Assessment, Development, and Evaluations (GRADE) method was used to evaluate the certainty of evidence. The random-effects model was used in meta-analysis, and data were analyzed using standardized mean differences and 95% confidence intervals.

**Results:** A total of 22 studies were included in this review, all of which were rated as having good methodological quality based on the PEDro scale. In the meta-analysis, nine of these studies involving 684 participants were included, and results showed that PA interventions have a positive effect on increasing total PA levels in CRC survivors ($Z = 2.79$, $p = 0.005$). Results of subgroup analysis revealed that supervised PA interventions ($Z = 2.82$, $p = 0.005$) and PA interventions with multiple intervention components ($Z = 3.06$, $p = 0.002$) effectively increased total PA levels for CRC survivors. In addition, research evidence suggests that daily as the frequency ($Z = 4.28$, $p < 0.001$), Moderate-to-vigorous physical activity (MVPA) as the intensity ($Z = 2.29$, $p = 0.022$), aerobic combined with resistance exercise as the type of PA ($Z = 4.19$, $p < 0.001$) is appropriate for increasing total PA levels in CRC survivors.

**Conclusions:** The findings of this review provide strong evidence supporting the positive role of PA interventions in improving total PA levels among CRC survivors. This study offers preliminary insights into the appropriate patterns of PA interventions (*e.g.*, frequency, intensity, type) for enhancing total PA levels in CRC survivors. However, further high-quality clinical trials are needed to determine the optimal timing, duration, and delivery methods of PA interventions to maximize their effectiveness in this population.

# INTRODUCTION

As a commonly diagnosed cancer around the world, both the incidence and mortality of colorectal cancer (CRC) are among the leading causes of cancer globally, respectively ranking third and second in the world (*Bray et al., 2024*). Researchers predicted that 3.2 million new cases of CRC will occur in 2040 (*Xi & Xu, 2021*). In China, the incidence of CRC is expected to become the second incidence rate and the fifth mortality (*Xia et al., 2022*). With the popularization of early screening and advances in cancer therapy, the 5-year survival rate for CRC is increasing in China, similar to most developed European countries (*Wang et al., 2023*). Both the increasing incidence and survival rate of CRC suggest that China will have more individuals with CRC for an extended period, namely CRC survivors.

The National Cancer Institute defines cancer survivors as individuals living in peace with cancer from the date of the diagnosis (*Denlinger et al., 2014*). In other words, the cancer survivors are the individuals with cancer-free conditions after the time of diagnosis (*Denlinger, Barsevick & Robinson, 2009*). Presently, surgery, chemotherapy, radiotherapy, and immunotherapy are the primary therapeutic options for CRC survivors (*Dekker et al., 2019*). With no exception, these treatments may result in a series of side effects such as excessive fatigue, cardiovascular or gastrointestinal toxicity, pain, physical function impairment, overweight, and even obesity (*Von Kemp & Cosyns, 2023*; *Najafizade, Ebrahimi & Hemati, 2023*; *Jereczek-Fossa, Marsiglia & Orecchia, 2002*). In addition, researchers have confirmed that increased body weight or obesity is strongly linked with increased overall mortality among cancer survivors (*Petrelli et al., 2021*; *Calle et al., 2003*). Thus, it is critical to investigate strategies for mitigating treatment-related symptoms, *e.g.*, overweight, in CRC survivors.

Increasingly high-quality evidence showed that regular and plentiful physical activity (PA) can significantly relieve the treatment-related side effects and improve the quality of life (QoL) among CRC survivors (*Eyl et al., 2020*; *Jung & Son, 2021*; *Geng et al., 2023*). The Survivorship Care Guideline suggests that CRC survivors should participate in PA for 150 min per week and incorporate strength training exercises in their daily lives at least twice a week (*El-Shami et al., 2015*). Despite these recommendations, compliance with the recommended PA guidelines in CRC survivors is abysmal, and many CRC survivors still show a low level of PA (*Kang et al., 2020*). For example, a recent cross-sectional study on the PA condition of 174 CRC survivors observed similar results: only 13 CRC survivors had a sufficient PA, whereas 56 CRC survivors were completely sedentary (*Bao et al., 2020*). Nevertheless, clinical implementations of PA intervention have shown promising outcomes in improving PA and reducing sedentary behavior for cancer survivors (*Coughlin, Caplan & Stone, 2020*; *Witlox et al., 2018*; *Hardcastle et al., 2023*). As indicated in a review, PA interventions using consumer wearable devices significantly increased PA

levels among CRC survivors (*Coughlin, Caplan & Stone, 2020*). Findings of a recent clinical trial on PA and physical fatigue among colon and breast cancer survivors found that such interventions have long-term positive effects on PA participation enhancement and fatigue reduction (*Witlox et al., 2018*). Other randomized controlled trials (RCTs) also highlighted the effectiveness of PA interventions in increasing PA levels and reducing sedentary behavior among cancer survivors (*Hardcastle et al., 2023*).

Indeed, the changes in individual health behavior (*e.g.*, PA) are influenced by various social, economic, and cultural factors. Research has shown that theory application facilitates the reveal of the multiple influencing factors for health behavior change and improves the success rate of the behavior change intervention (*Mbous, Patel & Kelly, 2020*). Therefore, theories of behavior change have been widely used in the design of PA intervention, and the common behavior change theory or model contains social cognitive theory (SCT), theory of planned behavior (TPB), and transtheoretical model (TTM) (*Leach et al., 2023*; *Van Blarigan et al., 2019*; *Van Waart et al., 2018*). However, few studies have systematically explored the theoretical framework of PA intervention design. In addition, existing systematic reviews have found that PA interventions significantly improve cancer survivors' QoL and mitigate their cancer-related fatigue (CRF) (*Geng et al., 2023*; *Machado et al., 2022*). Studies devoted to examining the effectiveness of PA interventions on improving PA levels in CRC survivors are still limited, and the appropriate patterns of PA for CRC survivors to increase their PA levels also are not identified in the current studies (*Jung & Son, 2021*; *Mbous, Patel & Kelly, 2020*). Research on the effectiveness of PA interventions and the appropriate patterns of PA is necessary for researchers to develop an effective PA intervention for CRC survivors with low levels of PA.

According to the exercise guideline for cancer survivors, this exercise guideline mainly introduced a series of exercise prescriptions based on the FITT principle (frequency, intensity, type, and time) for eliciting improvements in cancer-related health outcomes among cancer survivors, such as anxiety, depressive symptoms, fatigue, and so on (*Campbell et al., 2019*). The FITT principle is not only an essential basis for developing exercise prescriptions but also an important component in designing PA interventions (*Maddocks, 2020*). Specifically, the frequency of FITT principle refers to how often exercise or exercise session is conducted per week; the intensity of FITT is the intensity of each exercise or exercise session performed; the type of FITT is the modality of exercise, including aerobic exercise, resistance exercise, and flexibility exercise; the time of FITT is the duration of the exercise or exercise sessions, which is often measured in minutes or hours (*Maddocks, 2020*). Nevertheless, it remains challenging for CRC survivors with low PA levels to choose appropriate patterns of PA for improving their PA levels based on current evidence. Therefore, this review aims to (1) examine the effectiveness of PA interventions in improving PA levels, (2) explore the appropriate patterns of PA interventions for CRC survivors to improve their PA levels, and (3) provide future recommendations on developing PA interventions.

## METHODS

### Study design

This systematic review has been registered in the International Prospective Register of Systematic Reviews (PROSPERO) database (CRD42024531424). To enhance the rigor of reporting, this review conformed to the updated guidelines for the Preferred Reporting Items for Systematic Review and Meta-analysis (PRISMA) in 2020 (*Page et al., 2021*).

### Search strategy

To systematically search and integrate the existing evidence on PA interventions for CRC survivors, seven electronic databases were utilized to retrieve literature published between January 1, 2010, and March 1, 2024, including five English databases (Cochrane Library, Embase, PubMed, Scopus, and Web of Science) and two Chinese databases (China National Knowledge Infrastructure and Wan Fang Data). The date of January 1, 2010, was chosen as the beginning date for the search since it marks the publication of the initial PA guidelines targeted at cancer survivors (*Schmitz et al., 2010*). Each electronic database follows the principle of patient/population, intervention, comparison and outcomes (PICOS) and utilizes the subject terms combined with free terms to conduct a literature search. Furthermore, the database-specific search filters are also used in the literature retrieval to improve the quality of the final search output. The search terms included ('physical activity intervention' or 'physical activity' or 'exercise' or 'exercise intervention' or 'aerobic exercise' or 'resistance exercise' or 'flexibility exercise') AND ('colorectal cancer survivors' or 'colorectal cancer patients' or 'colorectal cancer' or 'colon cancer' or 'rectal cancer') AND ('randomized controlled trials'). The detailed search strategies utilized in the seven databases are shown in Table S1. In addition, a manual screening was conducted in the reference lists of the included studies to guarantee thorough coverage of the eligible literature.

### Eligibility criteria

The criteria for inclusion were as follows: (a) the targeted research population was the adult CRC survivors (≥18 years old); (b) PA is defined as any bodily movement that is generated by skeletal muscles and results in energy consumption (*Caspersen, Powell & Christenson, 1985*). Eligible interventions contained all forms of PA, and we did not impose any restrictions on the frequency, intensity, type, and time of PA interventions; (c) comparison control group received only usual care, without any measures aimed to promote PA during the trial period; (d) research outcome measure are PA levels. PA levels indicate the intensity and duration of the PA performed by participants over a period of time and typically measured by self-reported questionnaire or objective pedometers (*Dowd et al., 2018*); (e) the study design is RCTs (only RCTs were included in this study due to the fact that RCTs are considered to be the ideal study design for systematic reviews assessing the effectiveness of interventions (*Barker et al., 2023*)). Exclusion criteria included studies that (a) were conference abstracts, literature reviews, or study protocols; (b) were published in languages other than English or Chinese; (c) combined PA

interventions with other interventions, *e.g.*, dietary intervention; (d) did not set up an appropriate comparison group.

## Study selection

EndNote 20 was used to eliminate duplicate citations from all retrieved studies. Two researchers (JYM and XKQ) independently conducted the literature search and subsequently assessed whether they met the selection criteria according to the titles, abstracts, and full texts. In cases of disagreement, a third independent researcher (CW) further evaluated the related disagreement, and consensus was reached through multiple discussions.

## Data extraction

Data extraction was conducted using a standardized form to extract relevant information from each eligible study, including general information (authors, year of publication, and country), study characteristics (study design, sample size, intervention, and control group), patient characteristics (sex and cancer category), intervention characteristics (contents, theory, compliance, and duration), PA characteristics (frequency, intensity, type, and time), primary outcome, measurement tool, and results. In addition, we followed the previously published research to conduct subgroup analyses based on the characteristics of PA interventions (*i.e.*, intervention characteristics and PA characteristics) in the included studies (*Geng et al., 2023*). The characteristics of the included studies and PA interventions are respectively presented in Tables 1 and 2.

## Quality assessment

The methodological quality of the included studies was assessed using the Physiotherapy Evidence Database (PEDro) scale, and this scale is increasingly used for assessing the methodological quality of clinical trials included in systematic reviews across physiotherapy, health, and medical areas (*Cashin & McAuley, 2020*). A total of 11 items, each rated as zero or one, constitute the PEDro scale. For the quality of included studies, a total score of less than three indicates "poor," four to five indicates "reasonable," six to eight indicates "good," and more than eight indicates "excellent" methodological quality (*Cashin & McAuley, 2020*). Given that the literature on PA interventions specifically for CRC survivors is still very limited, we did not exclude any studies based on their methodological quality. Two researchers (JYM and XKQ) independently assessed the methodological quality of the included studies, and any disagreement was discussed with third researcher (CW) until a consensus was reached. The detailed results of the quality assessment of the included studies are shown in Table 3.

## Meta-analysis

The R and RStudio were utilized to perform statistics analysis, and meta-analysis was carried out with the meta (a kind of R package) (*Balduzzi, Rücker & Schwarzer, 2019*). Referring to the previously published study (*Mbous, Patel & Kelly, 2020*), selecting full-scale RCTs and pilot RCTs to conduct a meta-analysis to evaluate the effectiveness of PA interventions in improving the total PA levels is reasonable. The total PA levels

**Table 1 The characteristics of the included studies.**

| Authors, year, country | Study design | Sample size & sex | Cancer category | Intervention group | Control group | Primary outcome | Measurement tools | Results |
|---|---|---|---|---|---|---|---|---|
| *Backman et al. (2014)* Sweden | Pilot RCT | IG (n = 39) CG (n = 38) (Female = 69, Male = 8) | CRC (n = 18) BC (n = 59) | Group-based supervised PA intervention | Usual care | PA levels | Study-specific PA questionnaire | There was a significant improvement in PA levels in the IG group (p = 0.016). |
| *Cadmus-Bertram et al. (2019)* America | Pilot RCT | IG (n = 26) (Male = 0, Female = 26) CG (n = 24) (Male = 2, Female = 22) | CRC (n = 5) (IG = 1, CG = 4) BC (n = 45) (IG = 25, CG = 20) | Technology-based PA intervention | Usual care | PA levels | ActiGraph | Participants in the IG significantly increased their PA levels by 69 ± 84 min/week *vs.* a 20 ± 71 min /week decrease in the CG (p = 0.002). |
| *Courneya et al. (2016)* Canada | RCT | IG (n = 136) (Male = 63, Female = 73) CG (n = 137) (Male = 63, Female = 74) | CRC (n = 273) (IG = 136, CG = 137) | Supervised PA intervention | Health education materials | PA levels | TPAQ | Compared with CG, the IG significantly increased PA levels (p = 0.002). |
| *Falz et al. (2023)* German | RCT | IG (n = 76) (Male = 45, Female = 31) CG (n = 72) (Male = 43, Female = 29) | CRC (n = 19) (IG = 10, CG = 9) BC (n = 84) (IG = 43, CG = 41) PC (n = 45) (IG = 23, CG = 22) | Technology-based PA intervention | Health education materials | PA levels | Activity parameters | The PA levels per week were not significantly different in IG and CG (p = 0.055). |
| *Golsteijn et al. (2023)* Netherlands | RCT | IG (n = 249) (Male = 212, Female = 37) CG (n = 229) (Male = 204, Female = 25) | CRC (n = 186), (IG = 100, CG = 86) PC (n = 292) (IG = 149, CG = 143) | Technology-based PA intervention | Usual care | PA levels | ActiGraph and SQUASH | The IG did not perform better than CG in PA levels at 16 weeks (p > 0.05). |
| *Golsteijn et al. (2018)* Netherlands | RCT | IG (n = 249) (Male = 212, Female = 37) CG (n = 229) (Male = 204, Female = 25) | CRC (n = 186 (IG = 100, CG = 86) PC (n = 292) (IG = 149, CG = 143) | Technology-based PA intervention | Usual care | PA levels | ActiGraph and SQUASH | At 3 and 6 months, the IG significantly increased the level of PA than CG (p < 0.05). |

| Authors, year, country | Study design | Sample size & sex | Cancer category | Intervention group | Control group | Primary outcome | Measurement tools | Results |
|---|---|---|---|---|---|---|---|---|
| *Hardcastle et al. (2023)* Australia | RCT | IG (n = 43) (Male = 5, Female = 38) CG (n = 44) (Male = 8, Female = 36) | CRC (n = 21) (IG = 9, CG = 12) BC (n = 66) (IG = 34, CG = 32) | Technology-based PA intervention | Health education materials | PA levels | ActiGraph | There was a significant improvement in PA levels in the IG than CG at 12 weeks (p = 0.007). |
| *Hardcastle et al. (2021)* Australia | RCT | IG (n = 34) (Male = 13, Female = 21) CG (n = 34) (Male = 21, Female = 13) | CRC (n = 53) (IG = 23, CG = 30) GC (n = 15) (IG = 11, CG = 4) | Technology-based PA intervention | Health education materials | PA levels | ActiGraph | The PA level was significantly higher in the IG compared with CG at 24 weeks following (p = 0.036). |
| *Irwin et al. (2017)* America | RCT | IG (n = 95) (Male = 28, Female = 67) CG (n = 91) (Male = 17, Female = 74) | CRC (n = 20) (IG = 12, CG = 8) BC (n = 99) (IG = 49, CG = 50) OC (n = 67) (IG = 34, CG = 33) | Group-based supervised PA intervention | Usual care | PA levels | Interview-administered PA questionnaire | The IG significantly increased PA levels compared with the CG (p < 0.05). |
| *Kim et al. (2019)* South Korea | RCT | IG (n = 37) (Male = 18, Female = 19) CG (n = 34) (Male = 17, Female = 17) | CRC (n = 71) (IG = 37, CG = 34) | Home-based PA intervention | Usual care | PA levels | GLTEQ | Significant improvement in PA levels in the IG (p < 0.001). |
| *Leach et al. (2023)* America | Pilot RCT | IG (n = 15) (Male = 6, Female = 9) CG (n = 14) (Male = 7, Female = 7) | CRC (n = 29) (IG = 15, CG = 14) | Technology-based PA intervention | Usual care | PA levels | Accelerometer and IPAQ short form | The IG significantly improved PA levels than CG (p < 0.05). |
| *Lee, Kim & Jeon (2018)* South Korea | Pilot RCT | IG (n = 38) (Male = 18, Female = 20) CG (n = 34) (Male = 17, Female = 17) | CRC (n = 72) (IG = 38, CG = 34) | Home-based PA intervention | Usual care | PA levels | GLTEQ | The PA levels significantly increased by 206.4 ± 260.6 min/week in the IG (p < 0.001). |

| | Table 1 (continued) | | | | | | | |
|---|---|---|---|---|---|---|---|---|
| Authors, year, country | Study design | Sample size & sex | Cancer category | Intervention group | Control group | Primary outcome | Measurement tools | Results |
| Lee et al. (2017) South Korea | RCT | IG (n = 62) (Male = 31, Female = 31) CG (n = 61) (Male = 28, Female = 33) | CRC (n = 123) (IG = 62, CG = 61) | Home-based PA intervention | Usual care | PA levels | GLTEQ | Significant improvement in PA levels in the IG (p < 0.05). |
| Ligibel et al. (2012) America | RCT | IG (n = 61) (Male = 5, Female = 56) CG (n = 60) (Male = 4, Female = 56) | CRC (n = 21) (IG = 11, CG = 10) BC (n = 100) (IG = 50, CG = 50) | Technology-based PA intervention | Usual care | PA levels | 7-Day PAR | The IG increased the level of PA by more than 100 vs. 22% in CG (p < 0.05). |
| Maxwell-Smith et al. (2019) Australia | RCT | IG (n = 34) (Male = 13, Female = 21) CG (n = 34) (Male = 21, Female = 13) | CRC (n = 53) (IG = 23, CG = 30) GC (n = 15) (IG = 11, CG = 4) | Technology-based PA intervention | Health education materials | PA levels | ActiGraph | The level of PA significantly increased by 45 min/week in IG and decreased by 21 min/week in CG (p < 0.05). |
| Mayer et al. (2018) America | RCT | IG (n = 144) (Male = 70, Female = 74) CG (n = 140) (Male = 67, Female = 73) | CRC (n = 284) (IG = 144, CG = 140) | Technology-based PA intervention | Health education materials | PA levels | GLTPAQ | No significant difference in PA levels between IG and CG at six months (p = 0.08). |
| Pinto et al. (2013) America | RCT | IG (n = 20) (Male = 8, Female = 12) CG (n = 26) (Male = 12, Female = 14) | CRC (n = 46) (IG = 20, CG = 26) | Home-based PA intervention | Usual care | PA levels | Accelerometer and 7-day PAR | The IG showed a significant increase in the level of PA compared with CG at 3 months (p < 0.05). |
| Van Blarigan et al. (2022) America | Pilot RCT | IG (n = 22) (Male = 8, Female = 14) CG (n = 22) (Male = 11, Female = 11) | CRC (n = 44) (IG = 22, CG = 22) | Technology-based PA intervention | Health education materials | PA levels | ActiGraph | No group difference in PA levels between IG and CG (p > 0.05). |
| Van Blarigan et al. (2019) America | Pilot RCT | IG (n = 20) (Male = 8, Female = 12) CG (n = 21) (Male = 9, Female = 12) | CRC (n = 41) (IG = 20, CG = 21) | Technology-based PA intervention | Health education materials | PA levels | ActiGraph | No group difference in PA levels between IG and CG (p > 0.05). |

| Authors, year, country | Study design | Sample size & sex | Cancer category | Intervention group | Control group | Primary outcome | Measurement tools | Results |
|---|---|---|---|---|---|---|---|---|
| *Van Vulpen et al. (2016)* Netherlands | RCT | IG ($n$ = 17) (Male = 10, Female = 7) CG ($n$ = 16) (Male = 11, Female = 5) | CRC ($n$ = 33) (IG = 17, CG = 16) | Group-based supervised PA intervention | Usual care | PA levels | SQUASH | The IG showed more activity in PA than CG at 18 weeks. |
| *Van Waart et al. (2018)* Netherland | Pilot RCT | Onco-Move ($n$ = 8) (Male = 5, Female = 3) OnTrack ($n$ = 7) (Male = 2, Female = 5) CG ($n$ = 8), (Male = 2, Female = 6) | CRC ($n$ = 23) (Onco-Move = 8, OnTrack = 7, CG = 8) | Onco-Move Home-based PA intervention OnTrack: Supervised PA intervention | Usual care | PA levels | PASE | Compared with CG, IG significantly increased PA levels after chemotherapy. |
| *Witlox et al. (2018)* Netherlands | RCT | IG ($n$ = 119) (Male = 10, Female = 109) CG ($n$ = 118) (Male = 11, Female = 107) | CRC ($n$ = 33) (IG = 17, CG = 16) BC ($n$ = 204) (IG = 102, CG = 102) | Supervised PA intervention | Usual care | PA levels | SQUASH | The IG reported significantly higher PA levels than CG 4 years post-baseline ($p$ < 0.05). |

**Notes:**

*Onco-Move* is a home-based, low-intensity, individualized, self-managed PA intervention; *OnTrack* is a moderate-to-high-intensity, supervised combined resistance and aerobic exercise program.

**Abbreviations:** *BC*, breast cancer; *CG*, control group; *CRC*, colorectal cancer; *GC*, gynecologic cancer; *GLTEQ*, Godin Leisure Time Physical Activity Questionnaire; *IG*, intervention group; *IPAQ*, International Physical Activity Questionnaire; *min*, minute; *OC*, other cancer; *PA*, physical activity; *PASE*, Physical Activity Scale for the Elderly; *PC*, prostate cancer; *RCT*, randomized controlled trial; *SQUASH*, short questionnaire to assess health-enhancing physical activity; *TPAQ*, Total Physical Activity Questionnaire; *7-Day PAR*, 7-Day Physical Activity Recal.

(primary outcome) were the continuous data measured by questionnaires or accelerometers, and the measurement tools for total PA levels varied in the included studies. Due to the measurement tools for total PA levels varied in the included studies, the standardized mean deviation (SMD) was adopted in the meta-analysis. When studies reported multiple time points for outcome measures, the data of total PA levels close to the end of the intervention was extracted. The researchers extracted the sample sizes, mean of the total PA levels, and standard deviation of the total PA levels from intervention and control groups to conduct subsequent meta-analyses. Results of the meta-analysis are presented by the SMD, 95% confidence interval (CI), effect size ($Z$), and $p$-value. As for the interpretation of the value for SMD, an SMD of 0.2 represents a small effect size, an SMD of 0.5 represents a moderate effect size, and an SMD of 0.8 represents a large effect size. Heterogeneity among the included studies was examined by the $\chi^2$ test. If there was no significant heterogeneity between each study ($p > 0.10$, $I^2 < 50\%$), the fixed-effect model

**Table 2 The characteristics of the PA interventions among the included studies.**

| Authors, year | Intervention details | | | | PA details | | | |
|---|---|---|---|---|---|---|---|---|
| | Intervention contents | Intervention theory | Intervention compliance | Intervention duration | Frequency | Intensity | Type | Time |
| *Backman et al. (2014)* | Group-based supervised PA intervention<br>– Daily walks of 10,000 steps<br>– Weekly supervised group walk with 1 h | NR | Using a pedometer to track the number of steps per day (91%) | 10 weeks | Daily | NR | Aerobic exercise | 60 min |
| *Cadmus-Bertram et al. (2019)* | Technology-based PA intervention<br>– Using a Fitbit tracker to record daily steps and amount of PA<br>– Provide an educational handbook to inform the benefits of PA<br>– Setting in-person instructional and goal-setting sessions to help select appropriate individualized goals<br>– Providing social support to assist participants in achieving and maintaining their PA goals<br>– Setting email-based coaching to help participants set updated goals and provide suggestions<br>– Using electronic health record integration to assist clinicians in communicating with participants better | BCT | NR | 12 weeks | Daily | MVPA | Aerobic exercise | NR |
| *Courneya et al. (2016)* | Supervised PA intervention<br>– Setting clear and challenging exercise goals<br>– Providing some supervised exercise sessions<br>– Providing free or low-cost access to fitness facilities<br>– Frequent and ongoing contact with qualified staff<br>– Setting individual tailoring intervention<br>– Providing written materials and application of behavior modification techniques | TPB | Calculating the completion rates for supervised exercise sessions (68.2%) | 3 years | Determined by participants | Determined by participants | Aerobic exercise | NR |
| *Falz et al. (2023)* | Technology-based PA intervention<br>– Providing wearable devices for activity tracking and uploading<br>– Using the specific application to visualize the training video presentations, record heart rate, complete questionnaires, and receive PA feedback | NR | NR | 6 months | Twice a week | Determined by the perceived exertion | Resistance exercise | 30 min |
| *Golsteijn et al. (2023)* | Technology-based PA intervention<br>– Sending computer-tailored PA advice from a secure website on three occasions<br>– Providing a pedometer and access to interactive web-based content | BCT | NR | 16 weeks | NR | NR | NR | NR |
| *Golsteijn et al. (2018)* | Technology-based PA intervention<br>– Sending computer-tailored PA advice from a secure website on three occasions<br>– Providing a pedometer and access to interactive web-based content | BCT | NR | 16 weeks | NR | NR | NR | NR |

| Authors, year | Intervention details | | | | PA details | | | |
|---|---|---|---|---|---|---|---|---|
| | Intervention contents | Intervention theory | Intervention compliance | Intervention duration | Frequency | Intensity | Type | Time |
| *Hardcastle et al. (2023)* | Technology-based PA intervention<br>– Providing an intelligent tracker to display steps, distance, heart rate, and active minutes<br>– Setting telephone health coaching to support participants' self-efficacy and motivate increasing their PA levels | NR | NR | 12 weeks | NR | MPA | NR | NR |
| *Hardcastle et al. (2021)* | Technology-based PA intervention<br>– Providing a Fitbit Alta™ to record the amount of PA<br>– Setting two 2 h group sessions to provide PA recommendations, set PA goals, and build confidence<br>– A 20-min phone call during week-8 to provide support and feedback | NR | NR | 12 weeks | NR | MVPA | NR | NR |
| *Irwin et al. (2017)* | Group-based supervised PA intervention<br>– Twice-weekly supervised exercise sessions led by experienced trainers<br>– Providing a PA log book to record participants' exercise during and outside of the sessions | NR | Calculating the attendance for supervised exercise sessions (83%) | 12 weeks | Twice a week | Determined by participants | Aerobic and Resistance exercise | 90 min |
| *Kim et al. (2019)* | Home-based PA intervention<br>– Providing two types of exercise DVDs, which comprised 30 min of resistance training using major and core muscles to be performed at home per day<br>– Weekly meetings with exercise trainers in the clinic<br>– Providing weekly phone counsel or small group training sessions to help participants address problems and provide suggestions<br>– Sending daily text messages to check their completion of daily exercise | NR | NR | 12 weeks | Daily | MVPA | Aerobic and Resistance exercise | 30 min |
| *Leach et al. (2023)* | Technology-based PA intervention<br>– Twice-weekly exercise sessions were led by two study staff and held live on the Zoom application<br>– Five discussion sessions with approximately 30–45 min were held live on the Zoom application | SCT | Calculating the attendance for exercise sessions (85.8%) | 12 weeks | Twice a week | Determined by participant | Determined by participant | 60 min |
| *Lee, Kim & Jeon (2018)* | Home-based PA intervention<br>– Providing an exercise diary and pedometer to record the amount of PA<br>– Providing exercise videos that contained two 30-min resistance exercise programs<br>– Setting the counseling sessions to inform the benefits of PA<br>– Sending daily text messages or telephone counseling to check the completion of daily exercise | NR | NR | 6 weeks | Daily | MPA | Aerobic and Resistance exercise | 60 min |

(Continued)

| Authors, year | Intervention details | | | | PA details | | | |
| | Intervention contents | Intervention theory | Intervention compliance | Intervention duration | Frequency | Intensity | Type | Time |
|---|---|---|---|---|---|---|---|---|
| *Lee et al. (2017)* | Home-based PA intervention<br>– Providing an exercise diary and pedometer to record the amount of PA<br>– Providing DVDs that contained two 30-min resistance exercises using one's body weight<br>– Weekly telephone counseling sessions are held to check the PA logs and determine the appropriate intensity or duration of the exercise.<br>– Three clinic meetings to inform the benefits of PA for participants and teach them how to exercise correctly<br>– Sending daily text messages to remind participants to complete exercise tasks | NR | NR | 12 weeks | Daily | MPA | Aerobic and Resistance exercise | 60 min |
| *Ligibel et al. (2012)* | Technology-based PA intervention<br>– Semi-structured phone calls delivered by behavioral counselors to provide information regarding the importance of exercise and guidelines for exercise safety<br>– Providing a pedometer to record the number of minutes of exercise | SCT | NR | 16 weeks | Daily | MPA | Determined by participant | 30 min |
| *Maxwell-Smith et al. (2019)* | Technology-based PA intervention<br>– Using a Fitbit Alta to record daily steps and send automated prompts encouraging participants<br>– Two hours of group sessions facilitated by a behavior change specialist<br>– Setting 20-min telephone counseling during week 8 to provide support and feedback regarding PA progress and review goals | NR | NR | 12 weeks | Daily | MVPA | NR | NR |
| *Mayer et al. (2018)* | Technology-based PA intervention<br>– Using a smartphone application (SurvivorCHESS) to record PA<br>– Setting PA goals to ensure self-monitoring<br>– Utilizing social networking to provide social support<br>– Offering information on PA and health | SDT | NR | 6 months | Daily | NR | NR | 30 min |
| *Pinto et al. (2013)* | Home-based PA intervention<br>– Providing a home log to monitor PA participation and a pedometer to record daily steps and amount of PA<br>– Providing telephone counseling to strengthen the self-efficacy for exercise and set realistic outcome expectations<br>– Receiving a PA and CRC survivorship tip sheet each week | SCT TTM | NR | 12 weeks | Five times per week | MPA | Aerobic exercise | 30 min |

| Authors, year | Intervention details | | | | PA details | | | |
|---|---|---|---|---|---|---|---|---|
| | Intervention contents | Intervention theory | Intervention compliance | Intervention duration | Frequency | Intensity | Type | Time |
| *Van Blarigan et al. (2022)* | Technology-based PA intervention<br>- Providing a printed booklet to inform the importance of PA<br>- Sending daily text messages to remind participants to conduct exercise<br>- Using a Fitbit Flex 2 Fitness wristband to record PA<br>- Providing exercise applications and videos to help participants how to exercise at home correctly | TPB | Calculating the mean proportion of texts that requested a reply to each intervention participants responded (63%) | 12 weeks | NR | MVPA | Aerobic exercise, Resistance exercise and flexibility exercise | NR |
| *Van Blarigan et al. (2019)* | Technology-based PA intervention<br>- Providing printed educational materials to inform the importance of PA for cancer rehabilitation<br>- Using Fitbit Flex to record daily steps and amount of PA<br>- Sending daily text messages to remind participants | TPB | Using the Fitbit wear time and response rates to interactive text messages (74%) | 12 weeks | Two or three days per week | MVPA | Aerobic and Resistance exercise | NR |
| *Van Vulpen et al. (2016)* | Group-based supervised PA intervention<br>- Twice-weekly exercise sessions supervised by physiotherapists<br>- Providing an exercise log to record PA | SCT | Calculating the attendance percentage of the total offered classes (89%) | 18 weeks | Daily | Determined by individual preferences and fitness level | Aerobic and Resistance exercise | 30 min |
| *Van Waart et al. (2018)* | Onco-Move: Home-based PA intervention<br>- Encouraging participants to engage in at least 30 min of PA, 5 days per week, with an intensity level of 12–14 on the Borg scale of perceived exertion<br>- Receiving written information tailored to the individual's preparedness to exercise<br>- Providing activity diary to record PA<br>OnTrack: Supervised PA intervention<br>- Twice weekly supervised exercise sessions provided by physical therapists<br>- Providing activity diary to record PA | Onco-Move: TTM OnTrack: NR | Compliance with exercise programs was based on the number of attended sessions and the activity diary (Onco-Move: 100%; OnTrack: 61% and 71%) | 6 months | Onco-Move: Five times per week OnTrack: Five timesper week | Onco-Move:Level of 12–14 on the Borg scale of perceived exertion OnTrack: Level of 12–16 on the Borg scale of perceived exertion | Onco-Move: NR OnTrack: Aerobic and Resistance exercise | Onco-Move: 30 min OnTrack: 30 min |
| *Witlox et al. (2018)* | Supervised PA intervention<br>- Twice weekly supervised exercise sessions provided by a physiotherapist<br>- Weekly check the graphs and give positive feedback about the obtained results | SCT | NR | 18 weeks | Five times per week | Determined by individual preferences and fitness level | Aerobic and Resistance exercise | 30 min |

**Notes:**

Intervention theory is the theoretical basis for designing PA intervention; Intervention compliance informs the measure of compliance with PA intervention; *Onco-Move* is a home-based, low-intensity, individualized, self-managed physical activity program; *OnTrack* is a moderate-to-high intensity, supervised aerobic combined with resistance exercise program.
**Abbreviations:** *BCT*, behavioral change theory; *CHESS*, comprehensive health enhancement support system; *DVDs*, digital versatile discs; *NR*, not report; min, minute; *PA*, physical activity; *SCT*, social cognitive theory; *SDT*, self-determination theory; *TPB*, theory of planned behavior; *TTM*, transtheoretical model.

**Table 3 Quality assessment of the included studies.**

| Authors, year | Eligibility criteria | Randomized allocation | Hidden allocation | Baseline comparison between groups | Blind participants | Blind therapists | Blind assessors | Proper follow-up | Intention to treat analysis | Comparison between groups | Point estimate and variability | Total scores | Rank |
|---|---|---|---|---|---|---|---|---|---|---|---|---|---|
| Backman et al. (2014) | YES | YES | NO | YES | NO | NO | NO | YES | YES | YES | YES | 6 | Good |
| Cadmus-Bertram et al. (2019) | YES | YES | NO | YES | NO | NO | NO | YES | YES | YES | YES | 6 | Good |
| Courneya et al. (2016) | YES | YES | NO | YES | NO | NO | NO | YES | YES | YES | YES | 6 | Good |
| Falz et al. (2023) | YES | YES | NO | YES | NO | NO | NO | YES | YES | YES | YES | 6 | Good |
| Golsteijn et al. (2023) | YES | YES | NO | YES | NO | NO | NO | YES | YES | YES | YES | 6 | Good |
| Golsteijn et al. (2018) | YES | YES | NO | YES | NO | NO | NO | YES | YES | YES | YES | 6 | Good |
| Hardcastle et al. (2023) | YES | YES | YES | YES | YES | YES | YES | YES | YES | YES | YES | 10 | Excellent |
| Hardcastle et al. (2021) | YES | YES | YES | YES | NO | NO | YES | YES | YES | YES | YES | 8 | Good |
| Irwin et al. (2017) | YES | YES | NO | YES | NO | NO | YES | YES | YES | YES | YES | 7 | Good |
| Kim et al. (2019) | YES | YES | YES | YES | NO | NO | NO | YES | YES | YES | YES | 7 | Good |
| Leach et al. (2023) | YES | YES | YES | YES | NO | NO | YES | YES | YES | NO | YES | 7 | Good |
| Lee, Kim & Jeon (2018) | YES | YES | NO | YES | NO | NO | NO | YES | YES | YES | YES | 6 | Good |
| Lee et al. (2017) | YES | YES | NO | YES | NO | NO | NO | YES | YES | YES | YES | 6 | Good |
| Ligibel et al. (2012) | YES | YES | NO | YES | NO | NO | NO | YES | YES | YES | YES | 6 | Good |
| Maxwell-Smith et al. (2019) | YES | YES | YES | YES | NO | NO | YES | YES | YES | YES | YES | 8 | Good |
| Mayer et al. (2018) | YES | YES | NO | YES | NO | NO | NO | YES | YES | YES | YES | 6 | Good |
| Pinto et al. (2013) | YES | YES | NO | YES | NO | NO | YES | YES | YES | YES | YES | 7 | Good |
| Van Blarigan et al. (2022) | YES | YES | YES | YES | NO | YES | NO | YES | YES | YES | YES | 8 | Good |
| Van Blarigan et al. (2019) | YES | YES | YES | YES | NO | NO | NO | YES | YES | YES | YES | 7 | Good |
| Van Vulpen et al. (2016) | YES | YES | NO | YES | NO | NO | YES | YES | YES | YES | YES | 7 | Good |
| Van Waart et al. (2018) | YES | YES | YES | YES | NO | NO | NO | YES | YES | NO | YES | 6 | Good |
| Witlox et al. (2018) | YES | YES | NO | YES | NO | NO | NO | YES | YES | YES | YES | 6 | Good |

**Note:**
Total scores less than three indicates "*poor*," four to five indicates "*reasonable*," six to eight indicates "*good*," and more than eight indicates "*excellent*" methodological quality.

**Table 4 GRADE summary of evidence for PA interventions to CRC survivors.**

| No. of studies | Study design | Risk of bias | Inconsistency | Indirectness | Imprecision | Other considerations | PA interventions | Usual care | Relative (95% CI) | Absolute (95% CI) | Certainty |
|---|---|---|---|---|---|---|---|---|---|---|---|
| **Certainty assessment** | | | | | | | **No. of patients** | | **Effect** | | **Certainty** |
| **PA intervention vs. usual care—Total PA levels** | | | | | | | | | | | |
| 9 | Randomized trials | Serious[a] | Not serious | Not serious | Not serious | None | 345 | 339 | – | SMD 0.32 SD higher (0.09 higher to 0.54 higher) | ⊕⊕⊕◯ Moderate |
| **Supervised PA intervention vs. usual care—Total PA levels** | | | | | | | | | | | |
| 3 | Randomized trials | Serious[a] | Not serious | Not serious | Serious[c] | None | 116 | 117 | – | SMD 0.37 SD higher (0.11 higher to 0.63 higher) | ⊕⊕◯◯ Low |
| **PA intervention (≥ three intervention components) vs. usual care—Total PA levels** | | | | | | | | | | | |
| 5 | Randomized trials | Serious[a] | Not serious | Not serious | Not serious | None | 212 | 204 | – | SMD 0.47 SD higher (0.17 higher to 0.77 higher) | ⊕⊕⊕◯ Moderate |
| **PA intervention (Frequency = daily) vs. usual care—Total PA levels** | | | | | | | | | | | |
| 5 | Randomized trials | Serious[a] | Not serious | Not serious | Serious[c] | None | 151 | 146 | – | SMD 0.52 SD higher (0.08 higher to 0.97 higher) | ⊕⊕◯◯ Low |
| **PA intervention (Intensity = MVPA) vs. usual care—Total PA levels** | | | | | | | | | | | |
| 2 | Randomized trials | Serious | Not serious | Not serious | Serious[c] | None | 61 | 57 | – | SMD 0.53 SD higher (0.17 higher to 0.90 higher) | ⊕⊕◯◯ Low |
| **PA intervention (Type = aerobic combined with resistance exercise) vs. usual care—Total PA levels** | | | | | | | | | | | |
| 3 | Randomized trials | Serious[a] | Not serious | Not serious | Serious[c] | None | 81 | 76 | – | SMD 0.69 SD higher (0.37 higher to 1.02 higher) | ⊕⊕◯◯ Low |

**Notes:**
[a] Hidden allocation and blind methods not performed in the included studies.
[b] ≤100 participants.
[c] ≤400 participants.
**Abbreviations:** CI, confidence interval; PA, physical activity; SMD, standardized mean difference.

was used; if significant heterogeneity existed ($p < 0.10$, $I^2 > 50\%$), the random-effects model was used for meta-analysis. In addition, we used the sensitive analysis to explore the potential source of heterogeneity, and the egger test and funnel plot were utilized to evaluate publication bias in the included studies. The *p-value* less than 0.05 was considered to indicate a statistically significant difference.

## Evidence assessment

The Grading of Recommendations, Assessment, Development, and Evaluation (GRADE) method was used to assess the certainty of the effect of each outcome, which was significant in the meta-analysis (*Guyatt et al., 2008*). The assessment was conducted using the GRADEpro GDT software. Evidence based on RCTs was considered to be high-quality evidence at the outset, and certainty of evidence may be downgraded for several reasons,

including inconsistency of results, indirectness of evidence, imprecision, reporting bias, and study limitations (*Guyatt et al., 2008*). The certainty of evidence for the estimated effects can be categorized into four classes: high (very confident in the effect), moderate (moderately confident in the effect), low (little confidence in the effect), and very low (very little confidence in the effect) (*Guyatt et al., 2008*). The certainty of evidence was independently evaluated by one researcher (JYM) and confirmed by the second researcher (XKQ). The detailed results of the evidence assessment are shown in Table 4.

## RESULTS

### Study selection

The literature search initially yielded 1,986 potential studies from seven electronic databases, and two studies were identified through a manual search. After removing the duplicates, 1,230 articles remained. Then, 1,075 studies were removed because they did not meet the inclusion criteria by reading the title and abstract. Of these, 155 full-text articles were obtained for detailed eligibility assessment, and 133 studies that did not meet the eligibility criteria were excluded. Finally, 22 studies were included in this systematic review, and nine were selected for meta-analysis (see Fig. 1).

### Methodological quality of the included studies

The 22 studies included were evaluated for methodological quality following the 11 items of the PEDro scale. Of these, 21 studies demonstrated good methodological quality with total scores ranging from six to eight, while the remaining one study had excellent quality with a score of ten (*Hardcastle et al., 2023*). Each study stated its eligibility criteria, ensuring compliance with randomized allocation. Hidden allocation was employed in eight studies (*Hardcastle et al., 2023*; *Leach et al., 2023*; *Van Blarigan et al., 2019*; *Van Waart et al., 2018*; *Van Blarigan et al., 2022*; *Hardcastle et al., 2021*; *Maxwell-Smith et al., 2019*; *Kim et al., 2019*), while the remaining 14 did not report using this method (*Witlox et al., 2018*; *Golsteijn et al., 2023*; *Falz et al., 2023*; *Cadmus-Bertram et al., 2019*; *Mayer et al., 2018*; *Lee, Kim & Jeon, 2018*; *Golsteijn et al., 2018*; *Lee et al., 2017*; *Irwin et al., 2017*; *Van Vulpen et al., 2016*; *Backman et al., 2014*; *Pinto et al., 2013*; *Ligibel et al., 2012*; *Courneya et al., 2016*). Almost all included studies met the baseline comparison between groups, proper follow-up, comparisons between groups post-intervention, the point estimate, and variability of the PEDro scale. However, two studies did not meet the group comparisons after intervention (*Leach et al., 2023*; *Van Waart et al., 2018*). Despite this, these studies are still included in this review for subsequent analysis due to their good methodological quality and detailed statistics of interesting research outcomes. Additionally, the blindness of the participants, therapists, and assessors was the common reason that caused included studies to evaluate the "good" methodological quality. Specifically, one study reported that they blinded the participants in the research (*Hardcastle et al., 2023*), two studies suggested that they blinded the therapists (*Hardcastle et al., 2023*; *Van Blarigan et al., 2022*), and seven studies blinded the outcome assessors in their reports (*Hardcastle et al., 2023*; *Leach et al., 2023*; *Hardcastle et al., 2021*; *Maxwell-Smith et al., 2019*; *Irwin et al., 2017*; *Van Vulpen et al., 2016*; *Pinto et al., 2013*). In

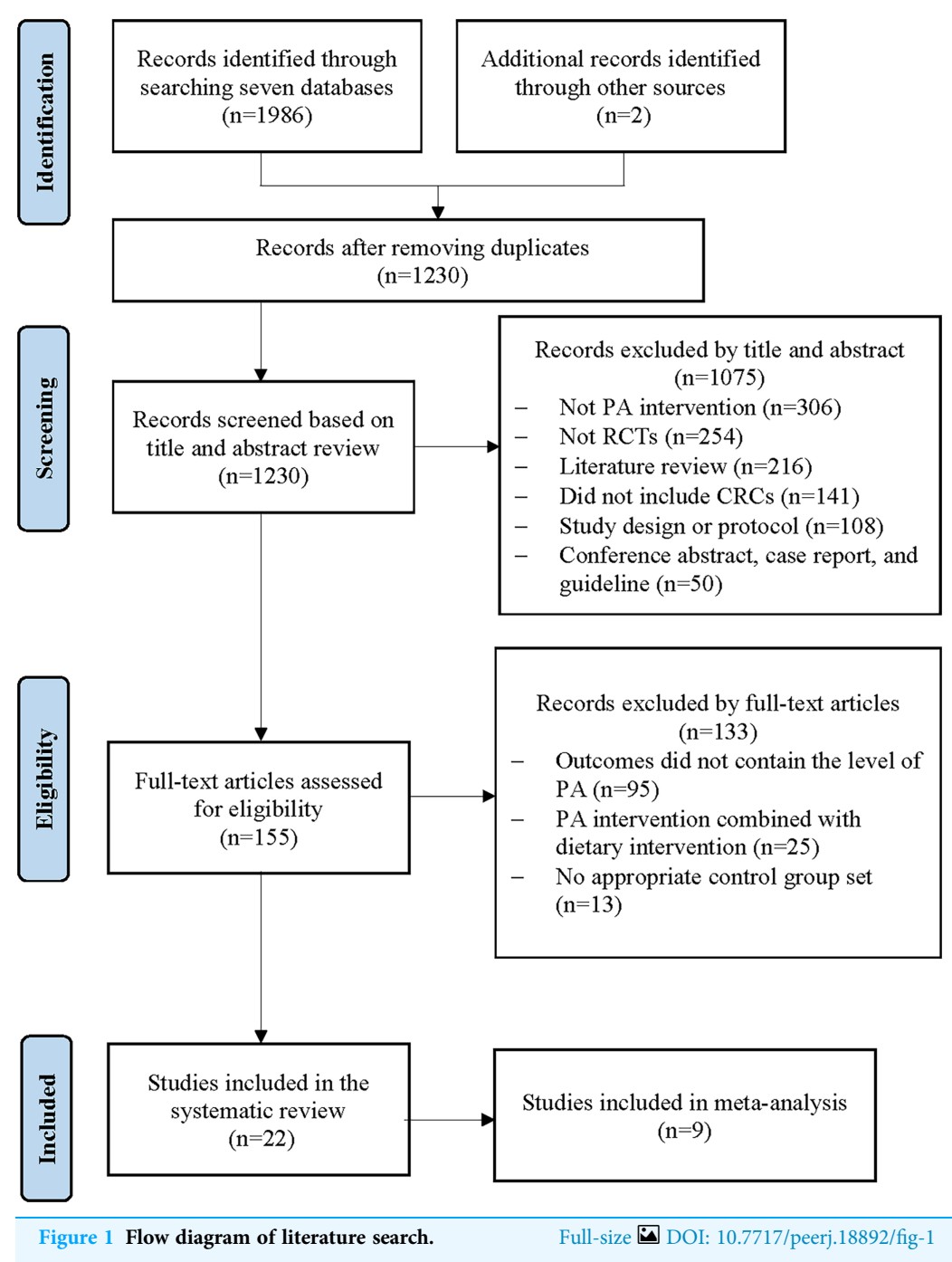

**Figure 1 Flow diagram of literature search.**

summary, the studies included in this review are characterized by good methodological quality.

## Study characteristics of the included studies

For general information, 15 studies were full-scale RCTs (*Witlox et al., 2018*; *Hardcastle et al., 2023, 2021*; *Maxwell-Smith et al., 2019*; *Kim et al., 2019*; *Golsteijn et al., 2023*; *Falz et al., 2023*; *Mayer et al., 2018*; *Golsteijn et al., 2018*; *Lee et al., 2017*; *Irwin et al., 2017*;

*Van Vulpen et al., 2016*; *Pinto et al., 2013*; *Ligibel et al., 2012*; *Courneya et al., 2016*), and the other seven were pilot RCTs (*Leach et al., 2023*; *Van Blarigan et al., 2019*; *Van Waart et al., 2018*; *Van Blarigan et al., 2022*; *Cadmus-Bertram et al., 2019*; *Lee, Kim & Jeon, 2018*; *Backman et al., 2014*). Of these 22 studies included, eight studies were from America (*Leach et al., 2023*; *Van Blarigan et al., 2019*, *2022*; *Cadmus-Bertram et al., 2019*; *Mayer et al., 2018*; *Irwin et al., 2017*; *Pinto et al., 2013*; *Ligibel et al., 2012*), five studies were from The Netherlands (*Witlox et al., 2018*; *Van Waart et al., 2018*; *Golsteijn et al., 2023*, *2018*; *Van Vulpen et al., 2016*), three studies were from Australia (*Hardcastle et al., 2023*, *2021*; *Maxwell-Smith et al., 2019*), three studies were from South Korea (*Kim et al., 2019*; *Lee, Kim & Jeon, 2018*; *Lee et al., 2017*), and the remaining three studies were from Canada (*Courneya et al., 2016*), Germany (*Falz et al., 2023*), and Sweden (*Backman et al., 2014*), respectively. The total sample size of the trials was 3,037, with each study consisting of 23 to 478 individuals. Each group had a balanced sample size and sex distribution in all the studies. In terms of the participant's cancer category, 11 studies only included CRC survivors (*Leach et al., 2023*; *Van Blarigan et al., 2019*; *Van Waart et al., 2018*; *Van Blarigan et al., 2022*; *Kim et al., 2019*; *Mayer et al., 2018*; *Lee, Kim & Jeon, 2018*; *Lee et al., 2017*; *Van Vulpen et al., 2016*; *Pinto et al., 2013*; *Courneya et al., 2016*), and the other 11 studies included mixed cancer survivors, including CRC, breast cancer, gynecologic cancer, and prostate cancer (*Witlox et al., 2018*; *Hardcastle et al., 2023*, *2021*; *Maxwell-Smith et al., 2019*; *Golsteijn et al., 2023*; *Falz et al., 2023*; *Cadmus-Bertram et al., 2019*; *Golsteijn et al., 2018*; *Irwin et al., 2017*; *Backman et al., 2014*; *Ligibel et al., 2012*).

Regarding the group characteristics, all selected studies were designed with appropriate research groups, including an intervention group and a control group. More than half of the included studies (12/22) for the intervention group utilized the technology-based PA interventions (*e.g.*, online training, computer-tailored PA advice, and video conference PA intervention) (*Hardcastle et al., 2023*; *Leach et al., 2023*; *Van Blarigan et al., 2019*, *2022*; *Hardcastle et al., 2021*; *Maxwell-Smith et al., 2019*; *Golsteijn et al., 2023*; *Falz et al., 2023*; *Cadmus-Bertram et al., 2019*; *Mayer et al., 2018*; *Golsteijn et al., 2018*; *Ligibel et al., 2012*), five studies utilized the supervised PA intervention (*Witlox et al., 2018*; *Irwin et al., 2017*; *Van Vulpen et al., 2016*; *Backman et al., 2014*; *Courneya et al., 2016*), four studies utilized the home-based PA intervention (*Kim et al., 2019*; *Lee, Kim & Jeon, 2018*; *Lee et al., 2017*; *Pinto et al., 2013*), and the remaining one study has two intervention arms, including the Onco-Move (*i.e.*, home-based, low-intensity, individualized, self-managed PA intervention) and OnTrack (*i.e.*, moderate-to-high intensity, supervised PA intervention) (*Van Waart et al., 2018*). The intervention contents of the different types of PA interventions are detailed in Table 2. For the control group, 14 studies showed that the control group only accepted usual care during the trial period (*Witlox et al., 2018*; *Leach et al., 2023*; *Van Waart et al., 2018*; *Kim et al., 2019*; *Golsteijn et al., 2023*; *Cadmus-Bertram et al., 2019*; *Lee, Kim & Jeon, 2018*; *Golsteijn et al., 2018*; *Lee et al., 2017*; *Irwin et al., 2017*; *Van Vulpen et al., 2016*; *Backman et al., 2014*; *Pinto et al., 2013*; *Ligibel et al., 2012*), and eight studies provided health education materials (*e.g.*, health booklets or guidebooks) to control group participants as part of usual care (*Hardcastle et al., 2023*;

*Van Blarigan et al., 2019, 2022; Hardcastle et al., 2021; Maxwell-Smith et al., 2019; Falz et al., 2023; Mayer et al., 2018; Courneya et al., 2016*).

The measurement of PA levels is commonly divided into subjective and objective measurements (*Dowd et al., 2018*). Specifically, subjective measurements mainly use questionnaires (*e.g.*, Godin Leisure-Time Exercise Questionnaire) to investigate the PA levels of participants over a period of time, and objective measurements use the motion receptors (*e.g.*, ActiGraph GTX3+ accelerometer) to record the PA among participants and calculate their PA levels (*Dowd et al., 2018*). Of these 22 studies included, 11 studies used subjective questionnaires or scales (*Witlox et al., 2018; Van Waart et al., 2018; Kim et al., 2019; Mayer et al., 2018; Lee, Kim & Jeon, 2018; Lee et al., 2017; Irwin et al., 2017; Van Vulpen et al., 2016; Backman et al., 2014; Ligibel et al., 2012; Courneya et al., 2016*), seven studies utilized objective measurement tools (*Hardcastle et al., 2023; Van Blarigan et al., 2019, 2022; Hardcastle et al., 2021; Maxwell-Smith et al., 2019; Falz et al., 2023; Cadmus-Bertram et al., 2019*), and the remaining four studies used subjective questionnaires combined with objective measurement tools to measure participants' PA level (*Leach et al., 2023; Golsteijn et al., 2023, 2018; Pinto et al., 2013*). Among 22 studies, the results of 17 studies showed that PA interventions significantly increased PA levels in the intervention group, and the difference between each group was statistically significant (*Witlox et al., 2018; Hardcastle et al., 2023; Leach et al., 2023; Van Waart et al., 2018; Hardcastle et al., 2021; Maxwell-Smith et al., 2019; Kim et al., 2019; Cadmus-Bertram et al., 2019; Lee, Kim & Jeon, 2018; Golsteijn et al., 2018; Lee et al., 2017; Irwin et al., 2017; Van Vulpen et al., 2016; Backman et al., 2014; Pinto et al., 2013; Ligibel et al., 2012; Courneya et al., 2016*). The remaining five studies suggested that PA interventions increased PA levels in the intervention group, with no statistically significant difference between each group at the end of the intervention (*Van Blarigan et al., 2019, 2022; Golsteijn et al., 2023; Falz et al., 2023; Mayer et al., 2018*).

## PA intervention characteristics of the included studies

Among the studies included, the intervention contents of nine studies were based on a single theoretical framework, including the SCT ($n = 4$), TPB ($n = 3$), TTM ($n = 1$), and self-determination theory ($n = 1$) (*Witlox et al., 2018; Leach et al., 2023; Van Blarigan et al., 2019; Van Waart et al., 2018; Van Blarigan et al., 2022; Mayer et al., 2018; Van Vulpen et al., 2016; Ligibel et al., 2012; Courneya et al., 2016*). The design of the intervention contents in the four studies was based on multiple theories, such as the combination of SCT and TTM which served as the theoretical basis for designing the intervention contents (*Golsteijn et al., 2023; Cadmus-Bertram et al., 2019; Golsteijn et al., 2018; Pinto et al., 2013*). In addition, the remaining 11 studies did not provide the theoretical framework for the design of PA intervention (*Hardcastle et al., 2023; Van Blarigan et al., 2022; Hardcastle et al., 2021; Maxwell-Smith et al., 2019; Kim et al., 2019; Falz et al., 2023; Lee, Kim & Jeon, 2018; Lee et al., 2017; Irwin et al., 2017; Backman et al., 2014; Courneya et al., 2016*). Eight studies were conducted on compliance with PA interventions, and compliance with PA interventions ranged from 61% to 100% in the included studies (*Leach et al., 2023; Van Blarigan et al., 2019; Van Waart et al., 2018; Van Blarigan et al., 2022; Irwin et al., 2017;*

*Van Vulpen et al., 2016; Backman et al., 2014; Courneya et al., 2016*). The intervention duration of included studies ranged from 6 weeks to 3 years, including the 6 weeks ($n = 1$) (*Lee, Kim & Jeon, 2018*), 10 weeks ($n = 1$) (*Backman et al., 2014*), 12 weeks ($n = 11$) (*Hardcastle et al., 2023; Leach et al., 2023; Van Blarigan et al., 2019, 2022; Hardcastle et al., 2021; Maxwell-Smith et al., 2019; Kim et al., 2019; Cadmus-Bertram et al., 2019; Lee, Kim & Jeon, 2018; Irwin et al., 2017; Pinto et al., 2013*), 16 weeks ($n = 3$) (*Golsteijn et al., 2023, 2018; Ligibel et al., 2012*), 18 weeks ($n = 2$) (*Witlox et al., 2018; Van Vulpen et al., 2016*), 6 months ($n = 3$) (*Van Waart et al., 2018; Falz et al., 2023; Mayer et al., 2018*), and 3 years ($n = 1$) (*Courneya et al., 2016*). The PA details (*i.e.*, frequency, intensity, type, and time of PA) are presented in Table 2.

## Overall analysis of the effect of PA interventions on total PA levels

Of the 22 studies reviewed, nine provided detailed statistics on total PA levels and were included in this meta-analysis (*Leach et al., 2023; Van Waart et al., 2018; Kim et al., 2019; Falz et al., 2023; Cadmus-Bertram et al., 2019; Lee, Kim & Jeon, 2018; Backman et al., 2014; Ligibel et al., 2012; Courneya et al., 2016*). Specifically, a study conducted by *Van Waart et al. (2018)* has two intervention arms: Onco-Move and OnTrack. These two intervention arms represent different types of PA interventions, so we divided the study into two parts for meta-analysis, following the approach outlined in the previously published literature (*Jung & Son, 2021*). Due to the different types of measurement tools used and significant heterogeneity ($p = 0.08$, $I^2 = 41\%$), we chose the SMD and the random-effects model to perform an overall meta-analysis of the effects of PA interventions on total PA levels (see Fig. 2). In the sensitivity analysis, the source of heterogeneity was not discovered. Results of the meta-analysis showed that PA interventions had a small-to-moderate positive effect on improving the total PA levels (SMD = 0.32, 95% CI [0.09–0.54], $Z = 2.79$, $p = 0.005$). In addition, according to the results of the egger test ($p = 0.42$) and funnel plot (see Fig. S1), there was no significant publication bias among the included studies.

## Subgroup analysis of the effect of PA interventions on total PA levels

A total of four studies using the technology-based PA intervention (*Leach et al., 2023; Falz et al., 2023; Cadmus-Bertram et al., 2019; Ligibel et al., 2012*), three studies using the home-based PA intervention (*Van Waart et al., 2018; Kim et al., 2019; Lee, Kim & Jeon, 2018*), and three using the supervised PA intervention were included for subgroup analysis (*Van Waart et al., 2018; Backman et al., 2014; Courneya et al., 2016*) (see Fig. 3). Random-effects model was used due to the significant heterogeneity ($p = 0.08$, $I^2 = 41\%$). Results of subgroup analysis showed that there were no statistically significant effects on total PA levels when the type of PA intervention was technology-based PA intervention (SMD = 0.15, 95% CI [−0.08 to 0.38], $Z = 1.27$, $p = 0.206$) and home-based PA intervention (SMD = 0.52, 95% CI [−0.06 to 1.11], $Z = 1.75$, $p = 0.080$). On the contrary, total PA levels were significantly increased when the type of PA intervention was supervised PA intervention (SMD = 0.37, 95% CI [0.11–0.63], $Z = 2.82$, $p = 0.005$).

A total of five studies with theory-based interventions (*Leach et al., 2023; Van Waart et al., 2018; Cadmus-Bertram et al., 2019; Ligibel et al., 2012; Courneya et al., 2016*) and five

| Study | Experimental Mean | SD | Total | Control Mean | SD | Total | Weight (common) | Weight (random) | Std. Mean Difference IV, Fixed + Random, 95% CI | Std. Mean Difference IV, Fixed + Random, 95% CI |
|---|---|---|---|---|---|---|---|---|---|---|
| Falz−2023 | 124.00 | 141.00 | 62 | 114.00 | 111.00 | 60 | 18.3% | 16.2% | 0.08 [−0.28, 0.43] | |
| Leach−2023 | 2679.50 | 330.60 | 13 | 3127.10 | 1768.40 | 12 | 3.7% | 6.1% | −0.35 [−1.14, 0.44] | |
| Cadmus−Bertram−2019 | 1463.00 | 489.00 | 24 | 1343.00 | 395.00 | 23 | 7.0% | 9.7% | 0.26 [−0.31, 0.84] | |
| Kim−2019 | 332.60 | 306.10 | 37 | 133.80 | 227.80 | 34 | 10.0% | 12.0% | 0.72 [0.24, 1.21] | |
| Lee−2018 | 571.10 | 382.40 | 38 | 309.40 | 196.70 | 34 | 9.9% | 11.9% | 0.84 [0.35, 1.32] | |
| Van Waart(1)−2018 | 97.70 | 38.00 | 7 | 120.60 | 46.00 | 8 | 2.2% | 3.9% | −0.51 [−1.54, 0.53] | |
| Van Waart(2)−2018 | 113.40 | 48.10 | 6 | 120.60 | 46.00 | 8 | 2.1% | 3.8% | −0.14 [−1.20, 0.92] | |
| Courneya−2016 | 1926.00 | 1842.00 | 106 | 1302.00 | 1212.00 | 105 | 31.1% | 19.5% | 0.40 [0.13, 0.67] | |
| Backman−2014 | 14.00 | 1.40 | 4 | 12.50 | 2.40 | 4 | 1.1% | 2.1% | 0.66 [−0.79, 2.12] | |
| Ligibel−2012 | 54.50 | 142.00 | 48 | 14.60 | 117.20 | 51 | 14.7% | 14.7% | 0.30 [−0.09, 0.70] | |
| **Total (fixed effect, 95% CI)** | | | **345** | | | **339** | **100.0%** | | **0.34 [0.19, 0.49]** | |
| **Total (random effect, 95% CI)** | | | | | | | | **100.0%** | **0.32 [0.09, 0.54]** | |

Heterogeneity: Tau² = 0.046; Chi² = 15.33, df = 9 (P = 0.0823); I² = 41%
Test for overall effect (fixed effect): Z = 4.35 (P < 0.0001)
Test for overall effect (random effects): Z = 2.79 (P = 0.0053)

−2   −1   0   1   2
Favours[control]   Favours[experimental]

**Figure 2  The effect of the PA interventions on total PA levels.** Note: *Falz et al. (2023)*, *Leach et al. (2023)*, *Cadmus-Bertram et al. (2019)*, *Kim et al. (2019)*, *Lee, Kim & Jeon (2018)*, *Van Waart et al. (2018)*, *Courneya et al. (2016)*, *Backman et al. (2014)*, *Ligibel et al. (2012)*.

studies with non-theory-based interventions (*Van Waart et al., 2018*; *Kim et al., 2019*; *Falz et al., 2023*; *Lee, Kim & Jeon, 2018*; *Backman et al., 2014*) were included for subgroup analysis (see Fig. 4). Random-effects model was used in both two subgroups due to the significant heterogeneity ($p = 0.08$, $I^2 = 41\%$). Results of subgroup analysis showed that theory-based PA interventions were not significantly effective in improving total PA levels (SMD = 0.23, 95% CI [−0.03 to 0.49], $Z = 1.75$, $p = 0.08$). In contrast, non-theory-based PA interventions had a significant effect in improving total PA levels (SMD = 0.46, 95% CI [0.04–0.87], $Z = 2.17$, $p = 0.03$).

A total of five studies with more than or equal to three intervention components (*Van Waart et al., 2018*; *Kim et al., 2019*; *Cadmus-Bertram et al., 2019*; *Lee, Kim & Jeon, 2018*; *Courneya et al., 2016*) and five studies with less than three intervention components (*Leach et al., 2023*; *Van Waart et al., 2018*; *Falz et al., 2023*; *Backman et al., 2014*; *Ligibel et al., 2012*) were included for subgroup analysis (see Fig. 5). Random-effects model was used due to the significant heterogeneity ($p = 0.08$, $I^2 = 41\%$). Results of subgroup analysis showed that there was a statistically significant effect for total PA levels when the number of PA intervention components was more than or equal to three (SMD = 0.47, 95% CI [0.17–0.77], $Z = 3.06$, $p = 0.002$). In contrast, PA intervention with less than three intervention components was not significantly effective in improving total PA levels (SMD = 0.13, 95% CI [−0.11 to 0.37], $Z = 1.03$, $p = 0.302$).

The different frequencies of PA were employed in these nine included studies, including the daily ($n = 5$) (*Kim et al., 2019*; *Cadmus-Bertram et al., 2019*; *Lee, Kim & Jeon, 2018*; *Backman et al., 2014*; *Ligibel et al., 2012*), twice a week ($n = 2$) (*Leach et al., 2023*; *Falz et al., 2023*), five times per week ($n = 1$) (*Van Waart et al., 2018*), and determined by participants ($n = 1$) (*Courneya et al., 2016*). These studies were included for subgroup analysis to determine the appropriate frequency of PA in improving total PA levels (see Fig. 6). Random-effects model was used in these four subgroups due to the significant heterogeneity ($p = 0.08$, $I^2 = 41\%$). Results of subgroup analysis revealed that there were no statistically significant effects for total PA levels when the frequency of PA was Twice a

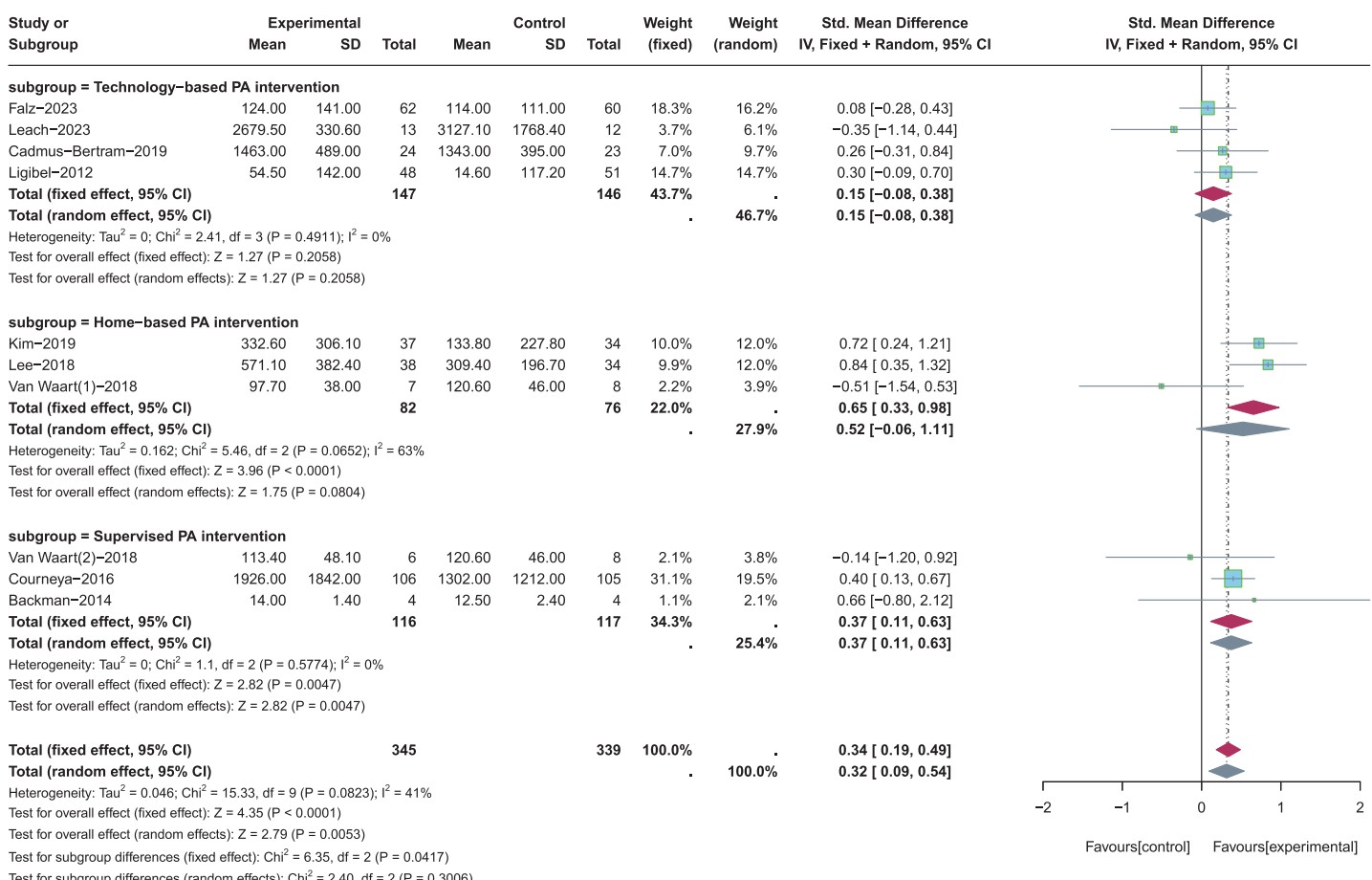

**Figure 3** The effect of intervention: the type of PA intervention. Note: *Falz et al. (2023)*, *Leach et al. (2023)*, *Cadmus-Bertram et al. (2019)*, *Kim et al. (2019)*, *Lee, Kim & Jeon (2018)*, *Van Waart et al. (2018)*, *Courneya et al. (2016)*, *Backman et al. (2014)*, *Ligibel et al. (2012)*.

week (SMD = 0.01, 95% CI [−0.32 to 0.33], Z = 0.04, *p* = 0.967) and five times per week (SMD = −0.33, 95% CI [−1.07 to 0.41], Z = −0.87, *p* = 0.383). Because there was only one study in the subgroup of "determined by participants", it was impossible to calculate the effect size of participant-determined PA frequency for improving total PA levels. In addition, total PA levels were significantly increased when the frequency of PA was daily (SMD = 0.53, 95% CI [0.29–0.77], Z = 4.28, *p* < 0.001).

The different intensities of PA were employed in these nine included studies, and one study did not report the intensity of PA in its research (*Backman et al., 2014*). Moderate-to-vigorous physical activity (MVPA) (*Kim et al., 2019*; *Cadmus-Bertram et al., 2019*), moderate physical activity (MPA) (*Lee, Kim & Jeon, 2018*; *Ligibel et al., 2012*), the intensity of PA determined by the perceived exertion (*Van Waart et al., 2018*; *Falz et al., 2023*), and the intensity of PA determined by participants (*Leach et al., 2023*; *Courneya et al., 2016*) were included for subgroup analysis to the determine the appropriate intensity of PA in improving total PA levels (see Fig. 7). Random-effects model was used in these four subgroups due to the significant heterogeneity (*p* = 0.06, $I^2$ = 47%). Results of subgroup analysis revealed that there were no statistically significant effects for total PA levels when

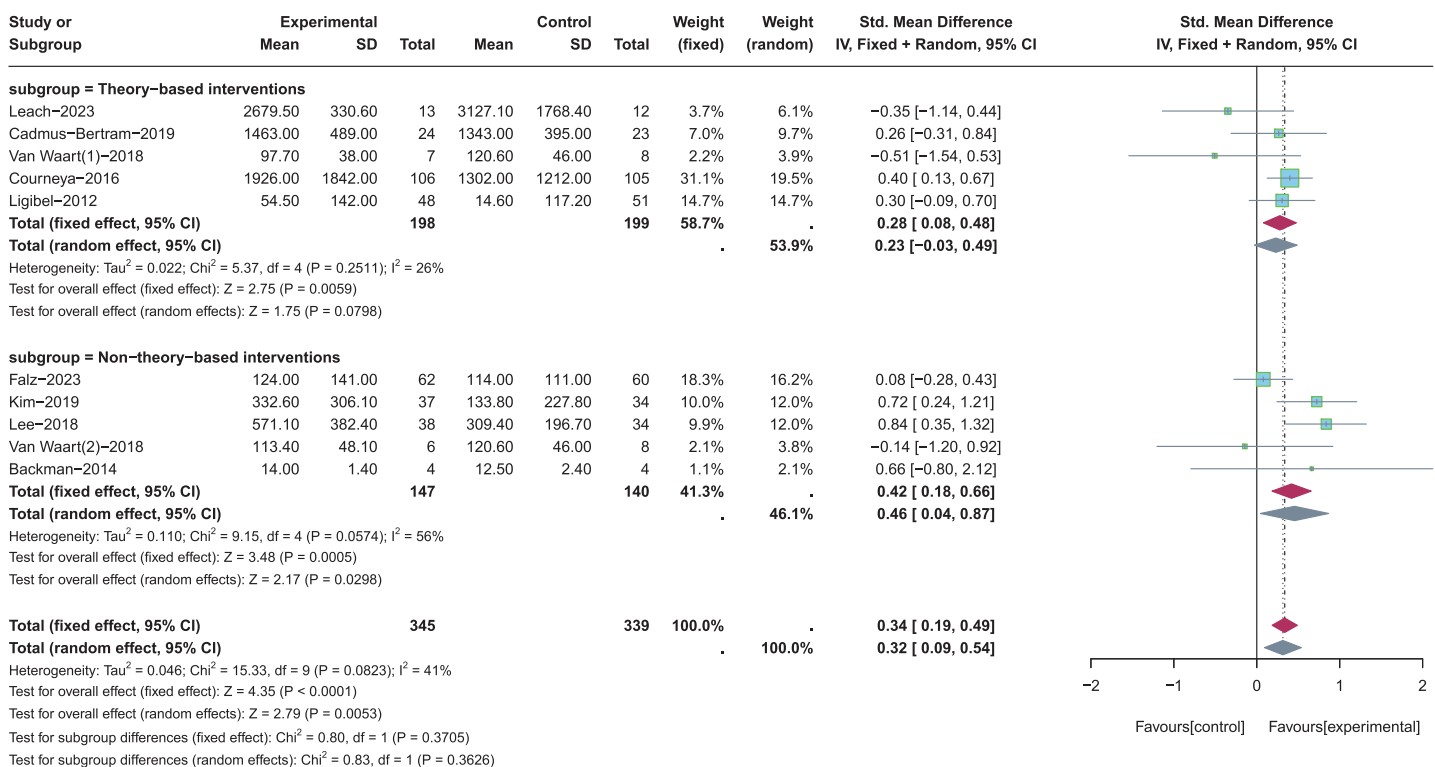

**Figure 4** **The effect of intervention: the theory application of PA intervention.** Note: *Falz et al. (2023)*, *Leach et al. (2023)*, *Cadmus-Bertram et al. (2019)*, *Kim et al. (2019)*, *Lee, Kim & Jeon (2018)*, *Van Waart et al. (2018)*, *Courneya et al. (2016)*, *Backman et al. (2014)*, *Ligibel et al. (2012)*.

the intensity of PA was determined by the perceived exertion (SMD = 0, 95% CI [−0.32 to 0.32], $Z = 0.01$, $p = 0.99$) or participants (SMD = 0.12, 95% CI [−0.58, 0.83], $Z = 0.34$, $p = 0.735$). On the contrary, total PA levels were significantly increased when the intensity of PA was MVPA (SMD = 0.52, 95% CI [0.08–0.97], $Z = 2.29$, $p = 0.022$) or MPA (SMD = 0.55, 95% CI [0.03, 1.07], $Z = 2.08$, $p = 0.038$). In summary, the pooled results showed that the intensity of PA was MVPA had more significant effects on increasing total PA levels than MPA ($Z = 2.29 > 2.08$, $p < 0.05$).

The different types of PA were employed in the nine included studies. Aerobic exercise (*Cadmus-Bertram et al., 2019*; *Backman et al., 2014*; *Courneya et al., 2016*), resistance exercise (*Falz et al., 2023*), aerobic combined with resistance exercise (*Van Waart et al., 2018*; *Kim et al., 2019*; *Lee, Kim & Jeon, 2018*), and type of PA determined by participants (*Leach et al., 2023*; *Ligibel et al., 2012*) were included for subgroup analysis to determine the appropriate type of PA in improving total PA levels (see Fig. 8). We utilized the fixed-effects model to perform the subgroup analysis as a result of the low heterogeneity ($p = 0.12$, $I^2 = 37\%$). Results of subgroup analysis revealed no statistically significant effects for total PA levels when the type of PA was determined by the participant (SMD = 0.17, 95% CI [−0.18 to 0.53], $Z = 0.96$, $p = 0.336$). Only one study was included in the subgroup of "resistance exercise" (*Falz et al., 2023*), so it was not possible to calculate the effect size. In contrast, aerobic exercise alone (SMD = 0.38, 95% CI [0.14–0.62], $Z = 3.08$, $p = 0.002$)

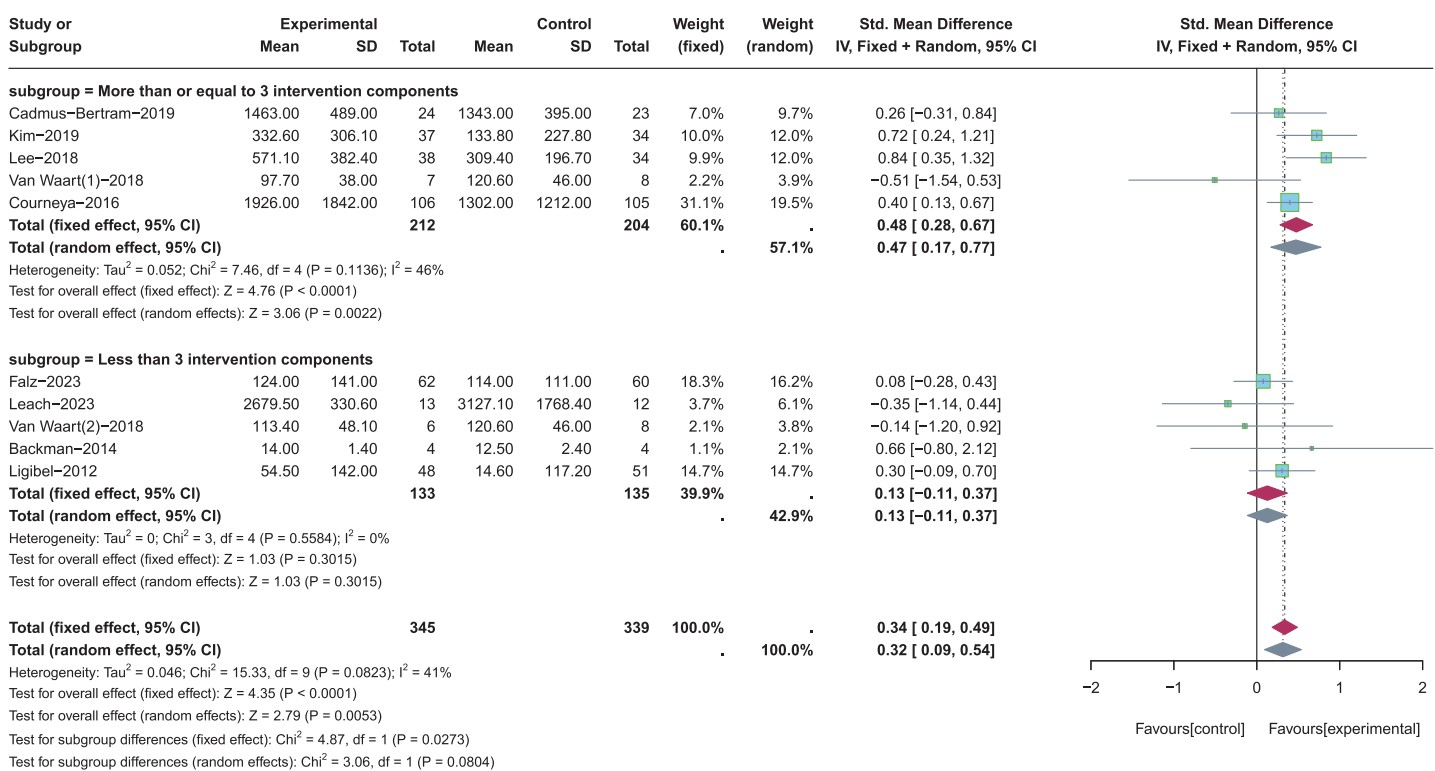

**Figure 5** **The effect of intervention: the number of PA intervention components.** Note: *Falz et al. (2023)*, *Leach et al. (2023)*, *Cadmus-Bertram et al. (2019)*, *Kim et al. (2019)*, *Lee, Kim & Jeon (2018)*, *Van Waart et al. (2018)*, *Courneya et al. (2016)*, *Backman et al. (2014)*, *Ligibel et al. (2012)*.

and particularly when combined with resistance exercise (SMD = 0.69, 95% CI [0.37, 1.02], $Z$ = 4.19, $p$ < 0.001) showed more substantial benefits for increasing total PA levels.

The different time of PA was employed in these nine included studies, and two studies did not report the time of PA in their research (*Cadmus-Bertram et al., 2019*; *Courneya et al., 2016*). The time of PA was 30 min (*Van Waart et al., 2018*; *Kim et al., 2019*; *Falz et al., 2023*; *Ligibel et al., 2012*) and 60 min (*Leach et al., 2023*; *Lee, Kim & Jeon, 2018*; *Backman et al., 2014*) were included for subgroup analysis to determine the appropriate time of PA in improving total PA levels (see Fig. 9). Random-effects model was used in the subgroup analysis due to the significant heterogeneity ($p$ = 0.04, $I^2$ =53%). Results of subgroup analysis revealed that there were no statistically significant effects for total PA levels when the time of PA was 30 min (SMD = 0.23, 95% CI [−0.10 to 0.56], $Z$ = 1.36, $p$ = 0.174) or 60 min (SMD = 0.38, 95% CI [−0.47 to 1.23], $Z$ = 0.87, $p$ = 0.382). In summary, the appropriate time of PA was not determined in this subgroup analysis due to the effect of each subgroup on improving the total PA levels is not significant.

## Certainty of evidence

The GRADE test evaluated the certainty of the effect for each significant outcome in the meta-analysis. Evidence on the effect of PA interventions on increasing total PA levels was rated as moderate quality because most of the included studies were at significant risk of

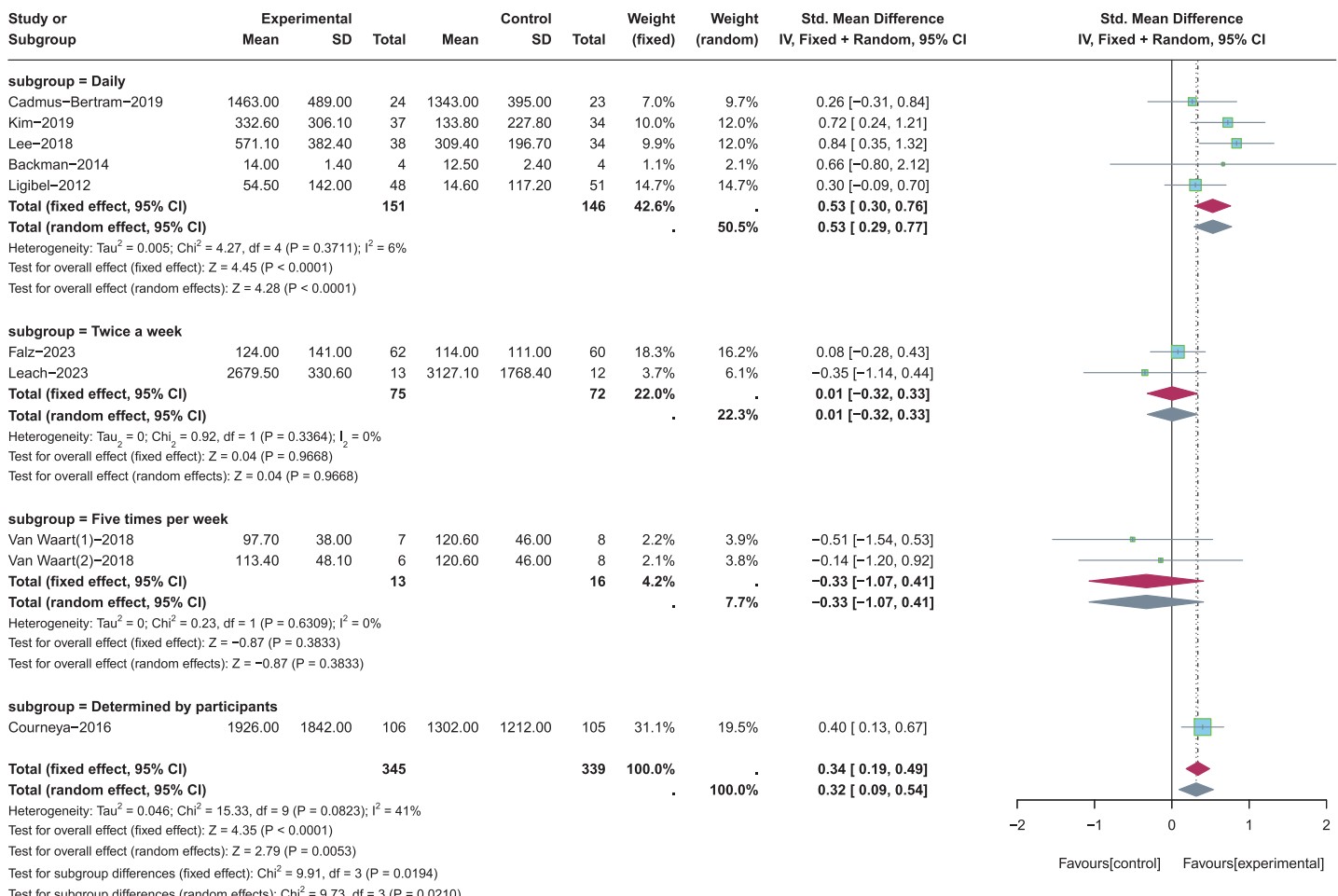

**Figure 6** **The effect of intervention: the frequency of PA.** Note: *Falz et al. (2023)*, *Leach et al. (2023)*, *Cadmus-Bertram et al. (2019)*, *Kim et al. (2019)*, *Lee, Kim & Jeon (2018)*, *Van Waart et al. (2018)*, *Courneya et al. (2016)*, *Backman et al. (2014)*, *Ligibel et al. (2012)*.

bias due to the lack of concealed allocation (*Falz et al., 2023*; *Cadmus-Bertram et al., 2019*; *Lee, Kim & Jeon, 2018*; *Backman et al., 2014*; *Ligibel et al., 2012*; *Courneya et al., 2016*) and the use of unblinded assessors (*Van Waart et al., 2018*; *Kim et al., 2019*; *Falz et al., 2023*; *Cadmus-Bertram et al., 2019*; *Lee, Kim & Jeon, 2018*; *Backman et al., 2014*; *Ligibel et al., 2012*; *Courneya et al., 2016*). For the characteristics of PA interventions, evidence on the effect of supervised PA intervention on increasing total PA levels was rated as low quality because most of the included studies presented serious risk of bias (*Van Waart et al., 2018*; *Backman et al., 2014*; *Courneya et al., 2016*) and imprecision. Evidence on the effect of PA intervention with more than three intervention components on increasing total PA levels was rated as moderate quality, and the serious risk of bias in the included studies was the main reason for downgrading this evidence (*Van Waart et al., 2018*; *Kim et al., 2019*; *Cadmus-Bertram et al., 2019*; *Lee, Kim & Jeon, 2018*; *Courneya et al., 2016*). For the type of PA, evidence on the effect of PA interventions (frequency = daily), PA intervention (intensity = MVPA), and PA intervention (type = aerobic combined with resistance

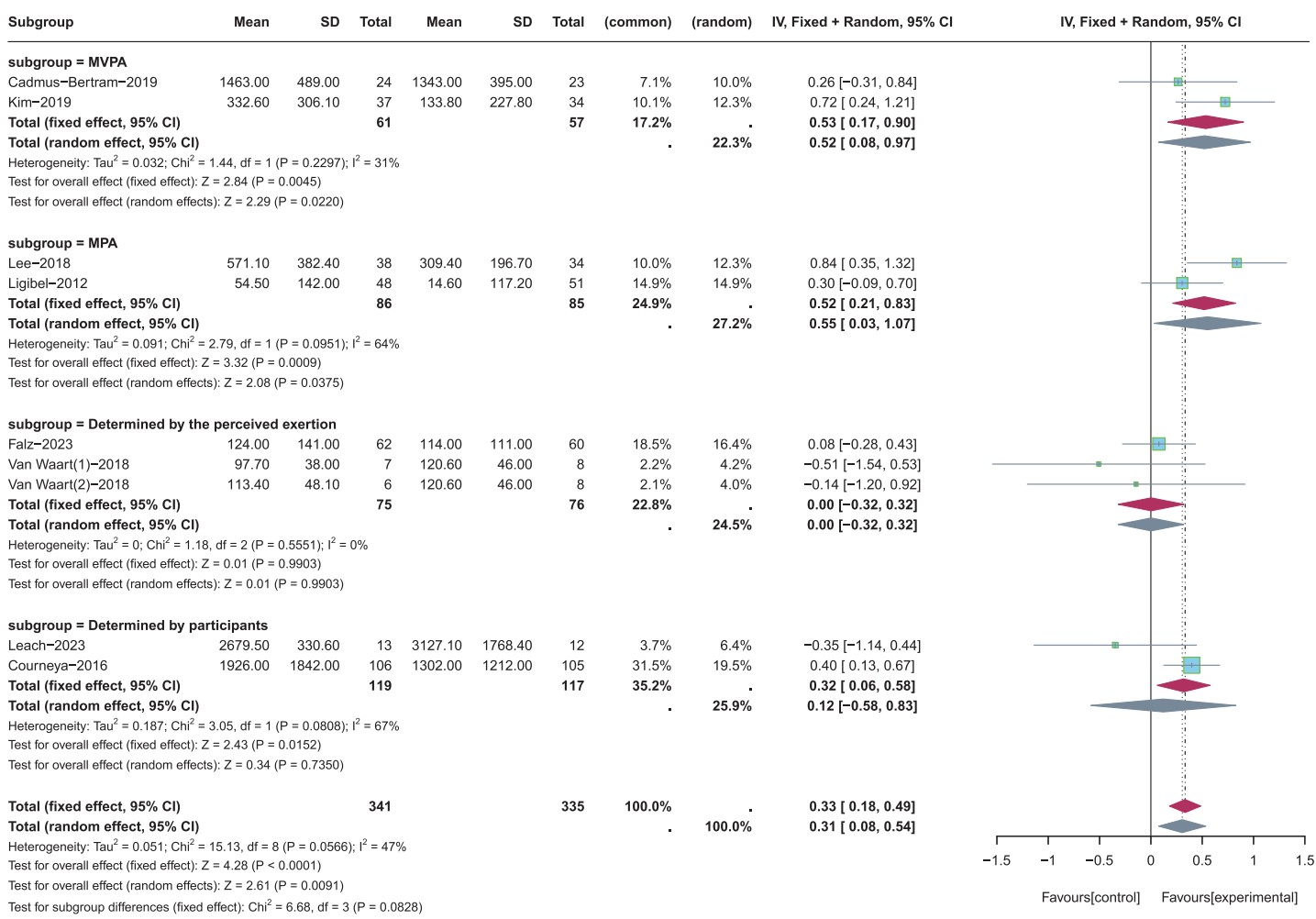

**Figure 7 The effect of intervention: the intensity of PA.** Note: *Falz et al. (2023)*, *Leach et al. (2023)*, *Cadmus-Bertram et al. (2019)*, *Kim et al. (2019)*, *Lee, Kim & Jeon (2018)*, *Van Waart et al. (2018)*, *Courneya et al. (2016)*, *Ligibel et al. (2012)*.

exercise) on increasing total PA levels was rated as low quality due to the serious risk of bias (*Kim et al., 2019*; *Cadmus-Bertram et al., 2019*; *Lee, Kim & Jeon, 2018*; *Backman et al., 2014*; *Ligibel et al., 2012*) and serious imprecision in the included studies. Therefore, a cautious interpretation and application of the above evidence is required due to the small sample size and high heterogeneity of the included studies.

## DISCUSSION

In summary, this study assessed the effectiveness of PA interventions for CRC survivors in improving their PA levels and aimed to explore the appropriate PA patterns for designing PA interventions for such populations. The results of meta-analysis showed that PA interventions have a small-to-moderate (SMD = 0.32, 95% CI [0.09–0.54], $Z = 2.79$, $p = 0.005$) effect on increasing the total PA levels for CRC survivors. Furthermore, a similar effect size was also observed in the previously published reviews (*Mbous, Patel & Kelly, 2020*; *Stacey et al., 2015*; *Groen, van Harten & Vallance, 2018*), and evidence on the effect of

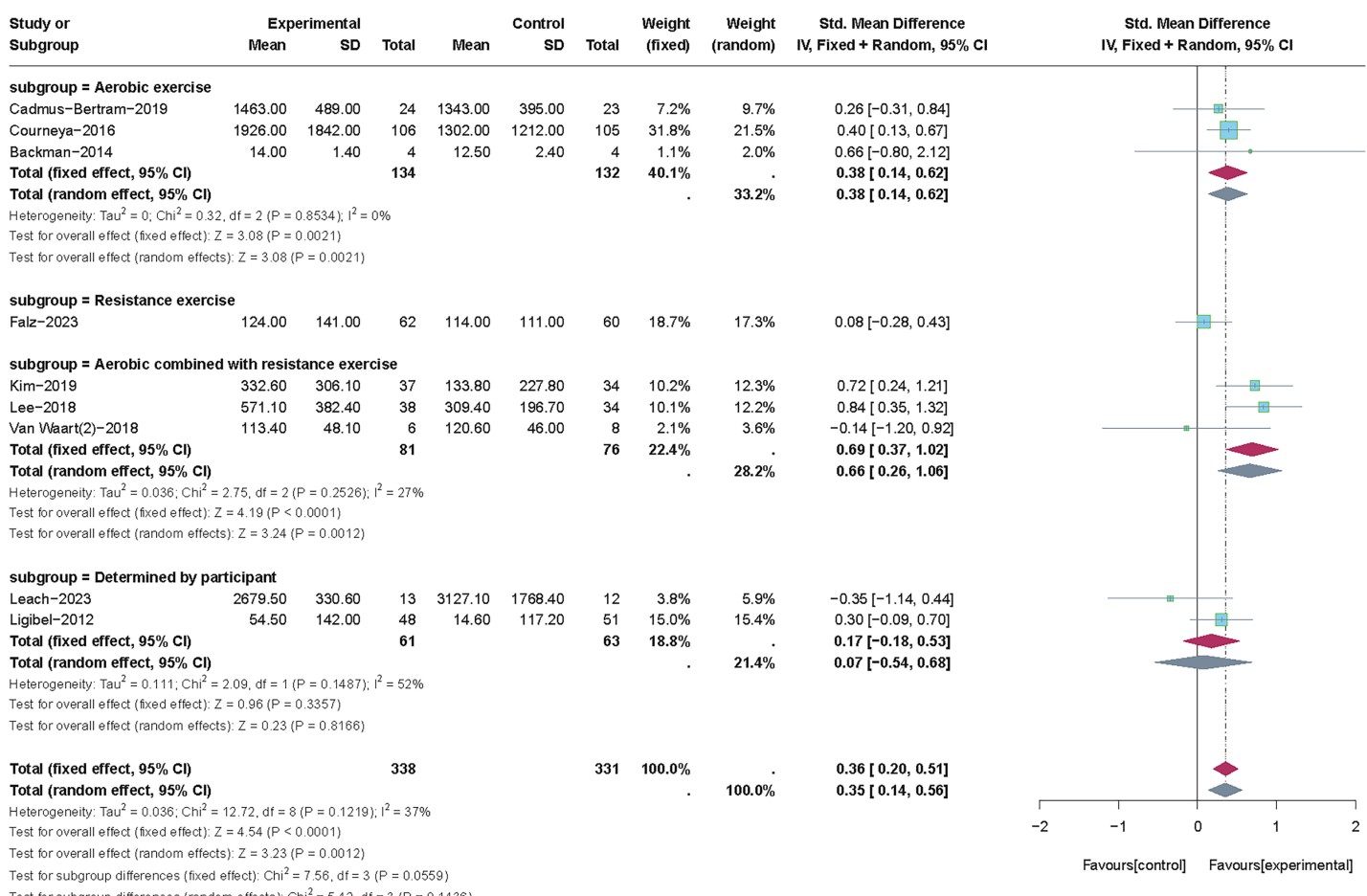

**Figure 8  The effect of intervention: the type of PA.** Note: *Falz et al. (2023)*, *Leach et al. (2023)*, *Cadmus-Bertram et al. (2019)*, *Kim et al. (2019)*, *Lee, Kim & Jeon (2018)*, *Van Waart et al. (2018)*, *Courneya et al. (2016)*, *Backman et al. (2014)*, *Ligibel et al. (2012)*.

PA interventions in increasing total PA levels was rated as moderate quality, suggesting that PA interventions are feasible for application in clinical practice for cancer survivors.

Although many clinical interventions have been developed to improve PA levels in CRC survivors, there is a lack of systematic reviews that have explored and synthesized these interventions. Existing evidence suggested that PA levels in CRC survivors were strongly associated with cancer-related symptoms and treatment-related side effects (*Kang et al., 2020*; *Fisher et al., 2016*). Conducting an effective PA intervention aiming at encouraging exercise and reducing sedentary behavior can enhance individuals' hemoglobin levels, and then their cardiopulmonary function, blood circulation, metabolism, and immune response improve, relieving cancer-related symptoms and treatment-related side effects (*Zhu et al., 2022*; *Hussey & Gupta, 2022*), which make CRC survivors actively engage in PA, ultimately increase their PA levels. In addition, PA interventions in the recent years have typically included components designed to enhance social interaction, including face-to-face or remote exercise sessions with physiotherapists *via* online platforms or smartphone applications (*Leach et al., 2023*; *Maxwell-Smith et al., 2019*). For instance,

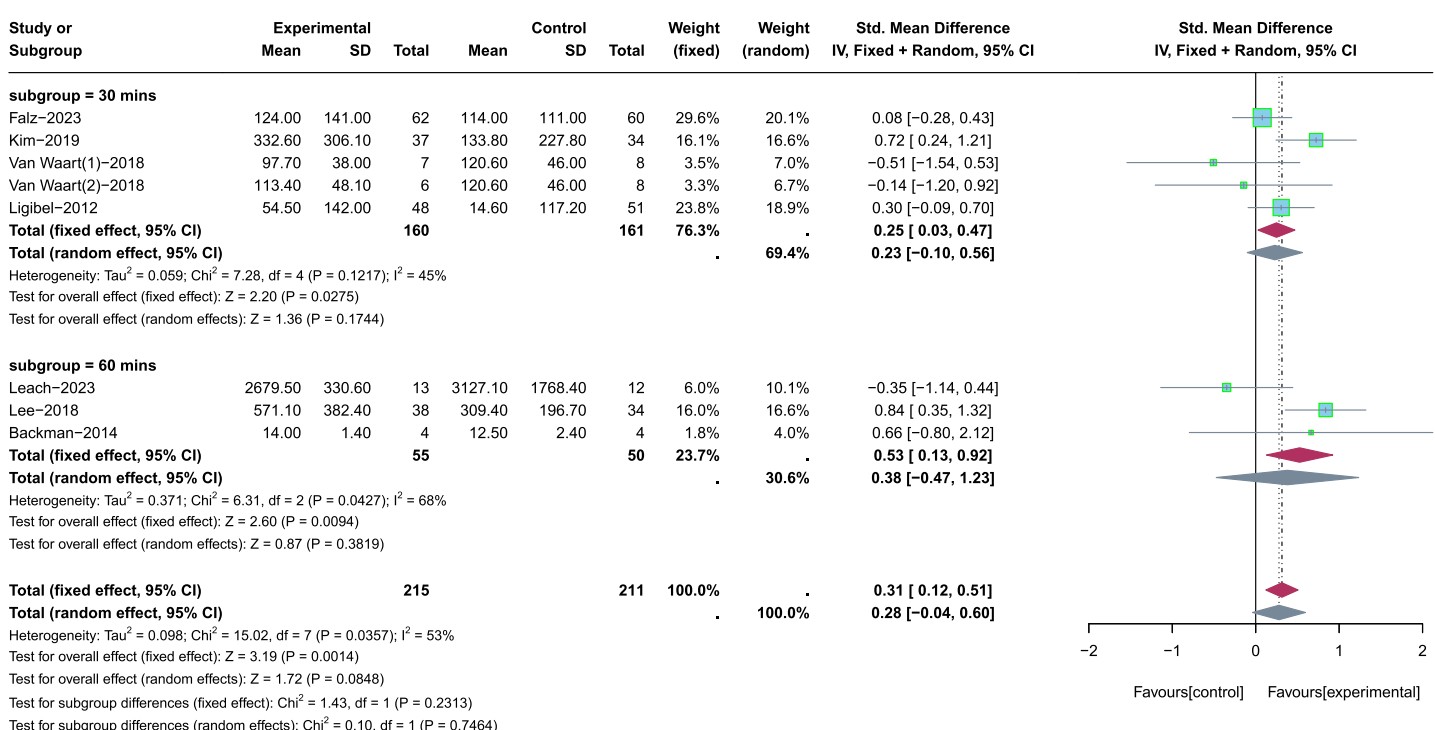

**Figure 9 The effect of intervention: the time of PA.** Note: *Falz et al. (2023)*, *Leach et al. (2023)*, *Kim et al. (2019)*, *Lee, Kim & Jeon (2018)*, *Van Waart et al. (2018)*, *Backman et al. (2014)*, *Ligibel et al. (2012)*.

*Leach et al. (2023)* combined traditional exercise with interactive Zoom sessions that guide CRC survivors through exercises and facilitate discussions with health professionals, enhancing their understanding and compliance with PA. Future studies should consider adding social interaction in the design of interventions to promote social inclusion for cancer survivors, aiming to enhance adherence to PA and reduce sedentary time.

The prognosis of CRC is improving with advancement in treatments, it is timely to consider the long-term effects of PA interventions. Long-term or short-term follow-up periods might influence the effect of PA interventions on increasing PA levels among CRC survivors. In this study, the follow-up periods of different types of PA interventions used for improving PA levels ranged from 6 weeks to 3 years, and 3 months ($n = 11$, 50%) was the most common follow-up period. This finding aligns with a recent study that found 3 months was the most common follow-up period for PA interventions exerting an effect on PA, and a significant improvement was observed at 6 months of follow-up (*McGettigan et al., 2020*). Therefore, choosing 3 months may be an appropriate follow-up time point to measure the effect of PA interventions, and clinical trials with long follow-up periods are still needed to validate the long-term effects of PA interventions. In addition, adherence to intervention is an essential factor affecting the maintenance of PA levels in cancer survivors after the intervention period. A total of eight studies included in this systematic review informed the compliance with PA interventions, and compliance with PA interventions ranged from 61% to 100%. Differences in adherence to the PA interventions may be due to differences in intervention contents in the included studies. For example, a

study conducted by *Van Waart et al. (2018)* has two intervention arms: Onco-Move and OnTrack, participants in the Onco-Move group ($n = 100\%$) had better adherence to the PA intervention than those in the OnTrack group ($n = 61\%$). A possible reason is that the Onco-Move intervention program developed the PA intervention based on the individual situation of the participants, and this intervention program is more suitable for participants, which is conducive to improving compliance with PA intervention (*Van Waart et al., 2018*). Therefore, future research is also needed to explore the current PA situation (*e.g.*, facilitators and barriers for PA participation) among CRC survivors and design tailored PA interventions for them to increase their adherence to these programs.

Regarding the characteristics of PA interventions, results of the meta-analysis showed that supervised PA interventions are more effective in increasing total PA levels among CRC survivors than technology-based or home-based PA interventions. This finding aligns with results from a clinical trial, which aimed to compare the effectiveness of home-based PA interventions with supervised PA interventions in improving PA levels for CRC survivors (*Van Waart et al., 2018*). Results of this trial suggested that for CRC survivors receiving chemotherapy, both home-based and supervised PA interventions are safe and feasible, but supervised PA interventions are more effective in increasing total PA levels (*Van Waart et al., 2018*). One possible reason is that supervised PA interventions have monitoring devices to provide participants with the supervision of experienced rehabilitation specialists or exercise physiologists during the exercise, significantly enhancing the safety of the intervention process and thereby improving participants' compliance with the intervention. This review also found that PA interventions with at least three intervention components can significantly increase total PA levels in CRC survivors. In these studies, the intervention components in the PA interventions ranged from two to six components, but most of the studies ($n = 5$, 55.6%) included in the meta-analysis only contained two intervention components in their PA intervention (*Leach et al., 2023*; *Van Waart et al., 2018*; *Falz et al., 2023*; *Backman et al., 2014*; *Ligibel et al., 2012*), and it may be an essential factor that affects the success rate of PA intervention. The most common components in PA intervention included holding supervised exercise sessions to teach PA, sending a pedometer or diary to monitor PA, and delivering daily phone calls or text messages to remind PA. However, most of the current reviews on PA intervention (including the present study) have not explored the optimal PA intervention components for increasing PA levels and improving other health-related outcomes (*Jung & Son, 2021*; *Geng et al., 2023*). Thus, a large number of high-quality clinical trials are still needed to explore the appropriate PA intervention components in the future. Furthermore, meta-analysis results showed that theory-based PA intervention did not significantly increase total PA levels in CRC survivors (SMD = 0.23, 95% CI [−0.03 to 0.49], $Z = 1.75$, $p = 0.08$). Our result is inconsistent with the previously published review, indicating that theory-based PA interventions successfully increased PA levels among CRC survivors (*Mbous, Patel & Kelly, 2020*). The reason for this discrepancy may be that most of the theory-based PA intervention studies included in this review only indicated their chosen intervention theory (*Van Waart et al., 2018*; *Ligibel et al., 2012*; *Courneya et al., 2016*) but did not integrate the theoretical framework with the design of the intervention
program, leading to ultimately undesirable intervention outcomes. Therefore, future research should carefully select behavior change theories and reasonably integrate the theoretical framework with intervention programs, which are critical for developing PA interventions to assist CRC survivors in improving their PA levels.

Regarding the characteristics of PA patterns, evidence from this review supports that daily appears to be an appropriate frequency of PA for increasing total PA levels in CRC survivors. This finding is consistent with the results of a previous study on home-based exercise programs, which suggested that daily as the frequency of PA could improve adherence to the intervention and increase PA levels in CRC survivors (*Lee, Kim & Jeon, 2018*). The reasons behind this phenomenon may lie in the fact that daily activities may be easily accepted by CRC survivors, such as daily walking or slow jogging, and a fixed frequency of PA can help participants remember and develop good PA habits. In addition, evidence from this review suggested that MVPA seems to be an appropriate intensity of PA for increasing total PA levels in CRC survivors. This finding is in line with a recent study that found the long-term performance of MVPA was positively associated with improvements in physical fatigue and sedentary time, thereby increasing PA levels in cancer survivors (*Mazzoni et al., 2023*). However, it has also been noted that the selection of exercise intensity in PA interventions needs to consider the previous exercise experience and current physiologic status of cancer patients and use the principle of progressive overload to gradually enhance exercise intensity during the intervention period (*Maddocks, 2020*). Therefore, conclusions reached in this review are more like a recommendation for most CRC survivors to improve their PA levels, and the actual selection of exercise intensity in PA interventions needs to be considered in the context of the participant's actual circumstances. Furthermore, evidence from this review identifies that aerobic combined with resistance exercise may be the appropriate type of PA for increasing total PA levels in CRC survivors, and this finding aligns with a recently published study (*Tanriverdi et al., 2023*). The results of this study showed that aerobic combined with resistance exercise assists cancer survivors in improving exercise capacity, relieving pain or fatigue, and enhancing health-related QoL (*Tanriverdi et al., 2023*). A large number of clinical studies have shown that cancer-related symptoms, particularly fatigue, are one of the major barriers to PA participation in CRC survivors (*Kang et al., 2020*; *Van Waart et al., 2018*). Since aerobic combined with resistance exercise could significantly alleviate cancer-related symptoms (*Geng et al., 2023*; *Tanriverdi et al., 2023*), this may be an important reason why combining aerobic and resistance exercise improves PA levels in CRC survivors. However, this review failed to identify the appropriate time of PA increasing PA levels in CRC survivors, and the effect of 30 or 60 min on improving the total PA levels is not significant in CRC survivors ($p > 0.05$). Existing evidence-based exercise prescriptions suggest that the appropriate time of a single session of exercise to improve health-related outcomes (*e.g.*, anxiety, fatigue, and health-related QoL) is approximately 30 to 60 min (*Campbell et al., 2019*). Determining the appropriate time of PA is critical to improving intervention adherence, such exercise time being too short may compromise the effectiveness of the intervention, and the participant may not tolerate time that is too long. Due to the small sample size and high heterogeneity of the included

studies, the appropriate time of PA for increasing PA levels in CRC survivors remains unknown, and additional large-scale RCTs are still necessary to explore the appropriate time of PA.

## Suggestions for future PA intervention studies

Based on the characteristics of the included PA intervention studies and the results of the meta-analysis, the following recommendations are provided to guide the development of future PA interventions:

(1) Conducted country: the vast majority of PA intervention studies in our study were conducted in the western countries, such as the United States and The Netherlands ($n = 13$, 60%), highlighting the need for future interventions to focus on the eastern countries, particularly China.

(2) Target population: this review determined the effectiveness of PA intervention in increasing total PA levels for CRC survivors, suggesting that future PA interventions could be applied to CRC survivors with low PA levels.

(3) Research design: most of the included studies have methodological quality problems since these included studies did not use hidden allocation ($n = 14$) and impose blinding on participants or assessors ($n = 21$). In order to ensure the methodological quality of research, it is important that future studies utilize more robust study designs (*e.g.*, RCTs) and adherence to the Consolidated Standards of Reporting Trials (CONSORT) checklist.

(4) Theory application: results of the meta-analysis showed that non-theory-based PA interventions have a better effect on increasing total PA levels in CRC survivors than theory-based PA interventions ($Z = 2.17 > 1.75$, $p < 0.05$). One possible reason is that theory-based PA intervention studies included in this review only indicated their chosen intervention theory (*Van Waart et al., 2018*; *Ligibel et al., 2012*; *Courneya et al., 2016*) but did not integrate the theoretical framework with the design of the intervention program. Thus, future research should carefully select behavior change theories and describe how the theoretical frameworks were integrated into the study design of PA interventions, providing a valuable reference for similar intervention studies.

(5) Intervention components: results of the meta-analysis showed that PA interventions with more than or equal to three intervention components have a better effect on increasing total PA levels in CRC survivors than PA interventions with less than three intervention components ($Z = 3.06 > 1.03$, $p < 0.05$). Compared to conventional PA interventions, PA interventions with multiple intervention components ($n \geq 3$) focus on the psychological condition of the participants and provide enough social support, thus contributing to improved intervention adherence. Therefore, it is recommended that future studies employ PA interventions with multiple intervention components ($n \geq 3$).

(6) Outcome measure: PA levels in half of the included studies ($n = 11$, 50%) were measured using self-report questionnaires or scales, which may have led to recall

bias. Although activity loggers or accelerometers are not the gold standard for the measurement of PA levels, but they are an objective and relatively reliable instrument that can be applied to future studies.

### Implications for clinical practice

It demonstrates that PA interventions can significantly improve PA levels for CRC survivors, and evidence on the effect of PA interventions on increasing total PA levels was rated as moderate quality. Recent research also confirmed that PA has a solid theoretical basis and strong evidence for reducing symptom burden and improving physical function in cancer patients, leading to improved health-related QoL (*Maddocks, 2020*). Moreover, implementing exercise programs during adjuvant treatment for CRC survivors can yield cost savings of 4,321 euros, highlighting the cost-effectiveness of such interventions (*May et al., 2017*). In clinical practice, CRC survivors typically face a wide range of disease-related and treatment-related challenges that often lead to a lack of PA during cancer survivorship (*Fisher et al., 2016*). Therefore, PA interventions should be recognized by clinical practitioners as an important and cost-effective strategy for improving PA levels and other health-related outcomes and should be implemented in the care of CRC survivors.

## STRENGTHS AND LIMITATIONS

It is acknowledged that there are several strengths in this review. Above all, seven electronic databases (five in English and two in Chinese) were selected for this review, and comprehensive searches were conducted to ensure a broad coverage of relevant literature, thereby maximizing the inclusion of eligible studies. Then, this review followed the PRISMA guidelines. Two authors independently screened and selected eligible studies, while data extraction was performed by a third reviewer. Any disagreements were resolved through multiple discussions with a third senior evaluator until a consensus was reached. In addition, this review not only confirms the effectiveness of PA intervention in increasing total PA levels for CRC survivors but also provides a series of recommendations for clinical practitioners to develop PA interventions for CRC survivors with low PA levels.

However, several limitations also existed in this review. First, this review aims to systematically summarize studies that applied the PA intervention to increase PA levels for CRC survivors. Given the limited availability of studies specifically discussing CRC survivors, we incorporated PA intervention studies targeting mixed cancer survivor populations that included CRC survivors. Second, inclusion of studies in this review was limited to those published in English and Chinese, and relevant studies in other languages may have been overlooked. Third, we chose to include both full-scale and pilot RCTs in the meta-analysis to evaluate the effectiveness of PA interventions. However, this approach may lead to an overestimation of the effect size for significant results in the meta-analysis. Fourth, apart from the overall analysis of the effect of PA interventions on total PA levels, most subgroup analyses have a small number of included studies and limited sample sizes, which could affect the robustness and reliability of the final findings.

## CONCLUSIONS

In conclusion, this systematic review and meta-analysis provided evidence that PA interventions have a small-to-moderate effect on increasing total PA levels in CRC survivors, and results of subgroup analyses showed that supervised PA interventions and PA interventions with multiple intervention components ($n \geq 3$) were effective in increasing total PA levels. There is preliminary evidence that daily as the frequency, MVPA as the intensity, and aerobic combined with resistance exercise as the type of PA is the appropriate PA patterns to improve total PA levels for CRC survivors. Additional RCTs with large sample sizes are still required to determine the appropriate time of PA for increasing PA levels in CRC survivors, and most findings reached in this review necessitate further methodologically reasonable investigation before establishing a definitive recommendation.

### Funding

The work was supported by the National Natural Science Foundation of China (No. 82172844). The funders had no role in study design, data collection and analysis, decision to publish, or preparation of the manuscript.

### Grant Disclosures

The following grant information was disclosed by the authors:
National Natural Science Foundation of China: 82172844.

### Competing Interests

The authors declare that they have no competing interests.

### Author Contributions

- Jiayu Mao conceived and designed the experiments, performed the experiments, analyzed the data, prepared figures and/or tables, authored or reviewed drafts of the article, and approved the final draft.
- Xiaoke Qiu conceived and designed the experiments, performed the experiments, analyzed the data, prepared figures and/or tables, authored or reviewed drafts of the article, and approved the final draft.
- Yi Zhang conceived and designed the experiments, performed the experiments, analyzed the data, prepared figures and/or tables, authored or reviewed drafts of the article, and approved the final draft.
- Can Wang conceived and designed the experiments, performed the experiments, analyzed the data, authored or reviewed drafts of the article, and approved the final draft.
- Xueli Yang conceived and designed the experiments, performed the experiments, authored or reviewed drafts of the article, and approved the final draft.

- Qiuping Li conceived and designed the experiments, performed the experiments, analyzed the data, prepared figures and/or tables, authored or reviewed drafts of the article, providing financial support, and approved the final draft.

## Data Availability

This is a systematic review/meta-analysis.

## Supplemental Information

Supplemental information for this article can be found online at http://dx.doi.org/10.7717/peerj.18892#supplemental-information.

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
