# Peer review of "A systematic review and meta-analysis of randomized controlled trials for physical activity among colorectal cancer survivors: directions for future research"

_PeerJ, doi:10.7717/peerj.18892_

## Round 0.1 · original submission · Major Revisions

Please thoroughly address all the points raised by the reviewers and have the manuscript edited by a native English speaker to ensure clarity and quality of scientific writing

Reviewer 1 ·

Basic reporting

no comment

Experimental design

no comment

Validity of the findings

Improving physical activity is a cost-effective strategy to enhance the quality of life for CRC survivors. However, guidelines on the frequency, intensity, type, and duration of physical activity remain unclear. This systematic review and meta-analysis investigate the effectiveness of physical activity interventions in promoting increased physical activity among CRC survivors. A total of nine clinical trials, involving 684 patients, were included in the meta-analysis.
Major concerns:
1. Subgroup analysis interpretation: The interpretation of certain subgroup analyses requires caution. For instance, the optimal duration of 1.5 months was derived from a single study. Extra care is needed in evaluating the significance of this result, taking into account factors such as exercise adherence, the total sample size, and other limitations. Reporting such findings as conclusive outcomes of a meta-analysis can be misleading.
2. Number of studies and sample size: Apart from the total physical activity analysis, most subgroup analyses have a small number of studies and limited sample sizes, which could affect the robustness and reliability of the findings.
Minor concerns:
1. Study representation: It is quite surprising that all nine included studies were conducted in developed countries, despite the inclusion of literature in both English and Chinese.
2. Wording clarity: Certain sentences could be improved for clarity. For example, “Subgroups were categorized as 'five days per week' (SMD = -0.33, 95% CI [-1.07, 0.41]), which indicated that PA interventions may be less effective than usual care in improving total PA levels.” This interpretation could be misleading due to the possibility of random errors. More precise wording would help to avoid overstatements.

·

Basic reporting

• The authors have made significant efforts to draft the manuscript in professional English. However, on several occasions it fails to meet the professional standards, and the manuscript will benefit from improvements in grammar, sentence structure and overall readability. Please conduct a thorough review to identify and fix the errors due to improper language throughout the manuscript.
• Current literature is appropriately cited throughout the manuscript, providing relevant context and background for the topics covered.
• The manuscript can benefit from providing a brief background on the topic “theories of behaviour change.” Since, the manuscript attempts to explore the theoretical framework for the physical activity intervention design, it is necessary to explain what the theories behind such intervention design. Alternatively, section 3.4.2 can provide a brief description of the theories included in this review.
• Line 96-98, the introduction of exercise guideline for cancer survivors needs correction. Please review reference number 26 to correctly identify the current exercise guidelines for cancer survivors and provide a brief summary relevant for the manuscript.
• Line 103, use the correct concepts for the FITT acronym. The authors have used the ‘time’ to indicate the “duration of the intervention.” The correct interpretation of ‘time’ is the duration of the exercise sessions, which is often measured in minutes or hours. This is a significant error and has been consistently repeated throughout the manuscript, please correct the use of the time aspect of FITT principle and clearly identify the duration of PA intervention with appropriate term. Correct the table 2S column title indicating “time.” Also, review line 314-318 for same issue.
• The manuscript has been formatted as per the Journal’s requirements and it conforms to acceptable formats. The multilevel numbering utilized in the manuscript can be reformatted to match the exact standards for the PeerJ requirements. The manuscript provided data in table and figure format.
• The manuscript uses the term “gender” (line 217) to refer to the biological sex of the participants, please consider using sex instead of gender. This is also the case for Table 1S.
• The table S2 of the supplementary file can be better placed between the main text in the manuscript. Additionally, since supplementary table 2S provides “characteristics of PA intervention,” I suggest that authors include all the relevant characteristics of the physical activity intervention included in the review. I suggest providing important characteristics of the PA intervention such as primary outcome measured, measurement tool, duration of PA intervention, measure of compliance with PA intervention among others. Additionally, the column title “Theory” does not convey the full scope of information captured, please add a footnote to explain that this provided the theoretical bases for the PA intervention design.
• Please see the following bullets for correction:
o Line 94, replace the term “inactive” with “sedentary” or “CRC survivors with low levels of PA.”
o Line 140-141, incorrect use of term “dependability, please revise it and write in full sentences for clarity.
o Line 149, replace “searching” with “search”
o Line 154, what does the authors mean by “features of research group?” Please clarify and explain how it is used in the systematic review and meta-analysis of subgroups.
o Line 155-156 is missing the work ‘respectively’
o Line 168-170 needs revision for clarify of message in the statement.
o Line 172, indicate mean and SMD for which outcome variable is used.
o Line 173, revise for clarity. Suggest reconsidering the word exhibited.
o Line 175, missing the word “effect size” at the end of sentence.
o Line 195, revise to improve clarity. “758 studies were found to be duplicate and were removed. For screening step titles and abstracts were reviewed and 1075 studies were deemed irrelevant…”
o Line 197, “…further excluded as they did not met the selection criteria.”
o Line 199-203, indicates a limitation of the current study and should be placed in the limitation. Alternatively, study selection criteria can be modified to include cancers other than colorectal cancer.
o Line 207, indicate score on what scale and what test is being described. i.e. PEDro tool.
o Line 212-215, avoid classification of countries based on the development index as it can be a heavy argued topic, unless it is relevant for the objective of the study. Simply, mention the studies and the countries where they were conducted. In the current form, it is misleading to not classify Canada, Germany and Sweden as developed countries.
o Line 217, replace gender with sex. Review Journal’s policy on the preference of term for indicating biological sex.
o Line 218, specify the types of cancer for the participants from the 11 studies that had survivors of cancer besides colorectal cancer.
o Line 225, incomplete sentence, please revise.
o Line 229, correct grammar for “the printed educational…”
o Line 235-237, provide a brief description of hat the GLTEQ is and how it measures PA levels. I suggest the authors include this category of information in the Table 2S and move that table in the main text for better understanding. Providing a brief explanation on the PA levels assessment tool can also be helpful.
o Line 243, please explain further by what the authors mean by “partially increase the PA levels.”
o Line 283-284, revise the sentence and indicate, 11 studies did not provide theoretical framework for the PA intervention design, which is different from authors statement in line 283-284.
o Line 304, replace demonstrate with indicate.
o Line 308, replace “to be clear” with specifically.

Experimental design

• For the large part the research question is well defined. The manuscript would benefit from having a clear definition or operational definition of ‘physical activity’ and the ‘level of physical activity.’ Since, this is the primary outcome against which the interventions are evaluated, it is necessary to explain what constitutes and improvement in physical activity levels. Also, provide a brief commentary on how the evaluation of physical activity levels are presented in the current literature. This may involve expanding on the use of accelerometer or pedometer for measuring movement.
• Next, one of the aims for the study as indicated on line 109 involves exploring “optimal FITT of PA intervention.” I recommend changing the term ‘optimal’ to more appropriate term, as the identification of optimal characteristics of physical activity intervention on such small and heterogeneous group of studies is not suitable, particularly when used for providing recommendations for exercise prescriptions which can have clinical significance. Identification of optimal frequency (among other characteristics) of PA intervention may require systematic testing through RCTs and subsequent meta-analysis of PA frequency on PA outcomes. Further, please identify the use of ‘optimal’ throughout the manuscript and replace with a more suitable term.
• Some of the studies included in the meta-analysis were pilot RCTs and may not be sufficiently powered or designed to evaluate PA intervention effectiveness. How did the authors address this issue of combining the findings from RCT and pilot RCTs. Does this have implications on the effect size outcome from the meta-analysis results? I recommend authors to review literature on meta-analyses to identify methods reported to address such issues.
• The research manuscript uses rigorous methods and high technical standards in designing and conducting the literature search and quality evaluation of the eligible studies. This has contributed to the understanding of implications for the conclusions drawn by the authors.
• Pertaining to the rigor of the current investigation, it is commendable that the authors have complied with the current PRISMA standards. They also attempted to search databases in Chinese language, besides English language database, for the literature search. Nonetheless, the methods section can benefit from some improvements. First, provide a clear search strategy with the terms and concepts used for the literature review. The use of validated search filters for common concepts like ‘cancer,’ ‘physical activity,’ ‘intervention/RCT’ has been increasing in the literature and may help improve the quality of search output. Did the authors consider using database specific search filters for the current manuscript?
• For the quality assessment section, authors utilized the PEDro scale for assessing the methodological qualities of studies. This scale is designed for studies in Physiotherapy Evidence Database. Most of the studies included in this review do not fall in the PED database. Please provide a rationale for selecting PEDro scale over other more commonly used quality assessment tools for RCTs, such as Cochrane Risk of Bias (ROB) 2.0 Tool or Checklist for Randomized Controlled Trials (JBI).
• Line 164, the manuscript is missing an explanation on how the quality assessment outcome of an RCT influenced its inclusion in the systematic review and meta-analysis outcome. Did the authors exclude any studies based on their quality?
• The heterogeneity assessment was performed using chi-square test. At what level of P-value the heterogeneity was considered significant? The authors indicated I2 >50%. P value of 0.10 is typically used for the test of heterogeneity because of the lack of power for the test and I2 >50% indicates high heterogeneity between studies.
• Line 184-190, how did the evidence assessment classifications (low, moderate, high) inform the meta-analysis outcome. Did the authors use the classification to draw conclusions on the meta-analysis outcomes? Was any study dropped from the meta-analysis due to low methodological quality?
• Overall, the research undertaken in this manuscript is relevant for advancing the knowledge on improving physical activity in colorectal cancer survivors. However, some issues with interpretation of results need to be reconsidered.

Validity of the findings

• The manuscript presents findings from a systematic literature review and meta-analysis. It is commendable that the authors have identified the relevant trials for the review and performed a strong data analysis and interpretations of the PA interventions tested in these trials.
• Line 249-276, information provided in the section 3.4.1 is more suitable in the tabular format. Consider expanding the table 2S column on “content of PA intervention” with a brief explanation of what intervention was provided to the intervention arm, instead of just listing the tools used. The text can be used to explain the grouping of the type of interventions.
• Line 271-272, please elaborate on what is Onco-Move and OnTract. The readers can not be expected to review the original paper, so it is essential that all the information relevant to understand the intervention needs to be provided with appropriate citation.
• Line 286, section 3.4.3, information provided here is provided in the Table 2S and can be better presented in a table to follow with the main text.
• Line 290, FITT principle – time dimension is used incorrectly. See previous comment indicating the correct use of “time” dimension. Additionally, provide an explanation for how time duration of PA session will be considered? (i.e. minutes / session of PA)
• Line 322-324, the authors divided a study into two parts due to multiple intervention arm. Could the authors provide a rationale for this decision? Also, provide a reference in literature where such method is used.
• Line 327, provide P-value for the heterogeneity test results. Also, provide them for all the remaining meta-analyses.
• The meta-analysis section of the manuscript needs improvement. First, for each subgroup analysis provide a clear group categories utilized before presenting the effect size. Additionally, explain what the authors mean by a specific category. For instance, line 345 says “weekly” frequency of PA, which does not convey clear message. Due to the incorrect interpretation of Time dimension on FITT principle, the authors have incorrectly classified studied into different PA intervention frequency and time variable. It is difficult to validate the results of subgroup meta-analysis on frequency of PA intervention. It is recommended that authors carefully review the classification of identified RCTs into appropriate categories.
• Line 365 and 376, incomplete sentence.
• Line 388, indicate the name of the test utilized. I.e. GRADE test. Also, provide an explanation on how this assessment was used in drawing conclusions for “optimal” FITT subgroup analysis.
• The Discussion section needs to be carefully revised to make the writing style coherent with expectation in scientific literature. Consider significant improvement in sentence structure. Due to multiplicity of errors, it is highly difficult to provide suggestion on edits to improve writing. This also makes it difficult to understand authors’ arguments. Some lines where improvement is required are: 396, 400, 404, 405, 407-41, 420-421,
• I would like the authors to re-examine the results from the meta-analysis of subgroups and revise the section 4.2. Additionally, use a suitable word instead of “optimal” since the current study is not suitable for determining optimal characteristics of PA interventions.
• For the section 4.3, suggestion for future PA intervention studies, provide a rationale for each suggestions and how does the current review/meta-analysis inform this suggestion.
• Line 523-524, is not identifying a strength mere a step in the systematic review process.
• Line 525-528, with the current state of evidence presented in this manuscript, the rigour and strength of evidence to identify optimal PA intervention is highly unlikely. Kindly, revise the statement to indicate more evidence / RCTs are necessary.
• Line 530-531 provides incorrect information. This review included survivors of other types of cancer besides colorectal cancer.
• Line 533-535 needs to be part of results section.

Additional comments

One of the more commonly asked question in the PA literature besides improving the level of PA is maintaining the improved level of PA obtained through intervention. Since, the prognosis of CRC is improving with advancement in treatments, it is timely to consider long-term effects of PA interventions. Stemming from this is the question of when was the outcome measured? Therefore, authors are recommended to provide a discussion on improved PA levels maintenance in survivors with temporal measurement of PA levels after the intervention period.

The authors should consider significant review of language, grammar, sentence structure and flow of ideas while drafting a manuscript. In the current state, it needs several revisions and it is difficult to identify all of them.

---

## Round 0.2 · accepted · Accept

The authors have satisfactorily addressed all the reviewers' comments, and the manuscript is now suitable for publication

Reviewer 1 ·

Basic reporting

The authors have addressed all the concerns raised in my initial review.

Experimental design

The authors have addressed all the concerns raised in my initial review.

Validity of the findings

The authors have addressed all the concerns raised in my initial review.

·

Basic reporting

none

Experimental design

none

Validity of the findings

none

Additional comments

The authors have made significant revisions to the manuscript. My comments have been sufficiently addressed and corresponding changes have been made in the revised manuscript. I congratulate the team for meticulously presenting the revisions and rebuttal.